

# Future methane fluxes of peatlands are controlled by management practices and fluctuations in hydrological conditions due to climatic variability

Vilna Tyystjärvi[1], Tiina Markkanen[1], Leif Backman[1], Maarit Raivonen[2], Antti Leppänen[2], Xuefei Li[2], Paavo Ojanen[3,4], Kari Minkkinen[3], Roosa Hautala[3], Mikko Peltoniemi[4], Jani Anttila[4], Raija Laiho[4], Annalea Lohila[1], Raisa Mäkipää[4], and Tuula Aalto[1]

[1]Climate System Research, Finnish Meteorological Institute, Helsinki, Finland
[2]Institute for Atmospheric and Earth System Research, Faculty of Science, University of Helsinki, Helsinki, Finland
[3]Department of Forest Sciences, University of Helsinki, Helsinki, Finland
[4]Natural Resources Institute Finland, Helsinki, Finland

**Correspondence:** Vilna Tyystjärvi (vilna.tyystjarvi@fmi.fi)

**Abstract.** Peatland management practices, such as drainage and restoration, have a strong effect on boreal peatland methane ($CH_4$) fluxes. Furthermore, $CH_4$ fluxes are strongly controlled by local environmental conditions, such as soil hydrology, tem-
perature and vegetation, which are all experiencing considerable changes due to climate change. Both management practices and climate change are expected to influence peatland $CH_4$ fluxes during this century but the magnitude and net impact of these changes is still insufficiently understood. In this study, we simulated the impacts of two forest management practices, rotational forestry and continuous cover forestry, as well as peatland restoration on hypothetical forestry-drained peatlands across Finland using the land surface model JSBACH coupled with the soil carbon model YASSO and peatland methane model
HIMMELI. We further simulated the impacts of climatic warming using two RCP (Representative Concentration Pathway) emission scenarios, RCP 2.6 and RCP 4.5. We investigated the response of $CH_4$ fluxes, soil water-table level (WTL), soil temperatures, and soil carbon dynamics to changes in management practices and climate. Our results show that management practices have a strong impact on peatland WTLs and $CH_4$ emissions continuing for several decades, with emissions increasing after restoration and clearcutting. Towards the end of the century, WTLs increase slightly likely due to increasing precipitation.
$CH_4$ fluxes have opposing trends in restored and drained peatlands. In restored peatlands, $CH_4$ emissions decrease towards the end of the century following the decomposition harvest residue in the top peat layers, while in drained peatland forests sinks get weaker and occasional emissions become more common, likely due to rising WTL and soil temperatures. The strength of these trends vary across the country, with $CH_4$ emissions from restored peatlands decreasing more strongly in southern Finland and forest soil $CH_4$ sinks weakening most in northern Finland.



## 1 Introduction

Boreal peatlands are considerable sinks of carbon and store approximately 270-370 Pg of carbon (Turunen et al., 2002). However, they are also a large source of methane ($CH_4$) (Turetsky et al., 2014; Abdalla et al., 2016), the second most important greenhouse gas after carbon dioxide (IPCC, 2013). In total, $CH_4$ emissions from northern peatlands form approximately 20 % of global wetland emissions and are a notable source of uncertainty in the global methane budget (Saunois et al., 2016).

The magnitude of $CH_4$ fluxes varies strongly depending on local factors, particularly soil water-table level (WTL) and soil temperature, controlling $CH_4$ emissions (Christensen et al., 2003; Turetsky et al., 2014). While pristine peatlands are strong sources of $CH_4$, drained and managed peatlands can turn to small sinks of $CH_4$ while simultaneously often turning to sources of $CO_2$ due to increased oxic soil respiration (Ojanen et al., 2010; Korkiakoski et al., 2019). On the other hand, restoration of drained peatlands to wetlands can reverse these changes, turning peatlands to sources of $CH_4$ and sinks of $CO_2$ (Wilson et al.,

2016). In order to understand these trade-offs and the role of $CH_4$ fluxes in the peatland carbon balance and the climate impacts of peatland management, it is important to study peatland $CH_4$ fluxes in managed peatlands for several decades after harvest and restoration managements.

$CH_4$ is produced in peatlands by microbes when soil carbon compounds decompose in anoxic conditions, typically in the water-logged peat layer (e.g. Lai, 2009). From the anoxic layer, $CH_4$ is transported to the atmosphere directly via plants and

through the above soil layers through diffusion and ebullition. Microbes also oxidise $CH_4$, largely in the oxic soil layer above water table (Xu et al., 2016). WTL is thus an important factor controlling the $CH_4$ flux from peatlands as it controls the thicknesses of both anoxic and oxic layers where $CH_4$ is produced and oxidised, respectively. However, previous studies have found that in wetlands where WTL stays constantly high, other factors become more important in controlling the variation of $CH_4$ flux (Olefeldt et al., 2013; Turetsky et al., 2014). Soil temperature has been found to be a significant controller of peatland

$CH_4$ flux, impacting soil microbial activity (Bubier et al., 1995; Zhao et al., 2016). Additionally, vegetation properties such as plant productivity and species composition control both methane production and transport (Dorodnikov et al., 2011; Turetsky et al., 2014).

While pristine peatlands are a source of $CH_4$, they are typically a net sink of carbon as soil organic matter gradually accumulates in the deep anoxic peat layers where decomposition is very slow (Turunen et al., 2002; Nilsson et al., 2008). However,

globally approximately 15 Mha, and 4.7 Mha in Finland, of boreal peatlands have been drained for forestry (Paavilainen and Päivänen, 1995; Päivänen and Hånell, 2012), which has a considerable impact on the ecosystem carbon balance (Ojanen et al., 2013; Korkiakoski et al., 2019; Mäkipää et al., 2023). The most common forest management option in peatland forests is rotational forestry which involves clearcutting the forest and requires ditches to control WTL in order to maintain sufficiently low WTL for forest production after forest harvesting (Paavilainen and Päivänen, 1995; Nieminen et al., 2018). While lowering

WTL decreases $CH_4$ emissions and may turn the soil to small $CH_4$ sinks (Ojanen et al., 2010), it simultaneously increases the soil $CO_2$ emissions by enhancing soil organic matter decomposition (Ojanen et al., 2013; Korkiakoski et al., 2019). This can turn the forest to a source of atmospheric $CO_2$ or decreasing its sink (Ojanen et al., 2010, 2013; Hommeltenberg et al., 2014), although this may not be the case in all nutrient-poor peatlands (Ojanen et al., 2013; Minkkinen et al., 2018). Furthermore,





clearcutting and ditch maintenance impose a heavy nutrient and carbon load on local water bodies, decreasing water quality
considerably (Nieminen, 2004). Therefore, management options that do not involve regular clearcuts or ditch maintenance
have been suggested in order to mitigate the climatic and environmental impacts of rotational forestry in boreal peatlands.

Continuous cover forestry has been suggested as an alternative forest management option to rotational forestry (Nieminen
et al., 2018). The precise harvesting methods vary but in effect only part of the forest stand is removed at one time, leading
to a heterogeneous forest structure. Continuous cover forestry has been shown to decrease the need to maintain ditches as the
continuous forest cover upholds a reasonably low WTL through evapotranspiration (Pothier et al., 2003; Leppä et al., 2020).
It can also improve peatland carbon balance as increased $CH_4$ and CO2 emissions following clearcut harvests can be avoided
(Korkiakoski et al., 2020, 2023).

To fully restore the climatic impacts and ecohydrological conditions of wetlands, restoration of drained peatlands by rewet-
ting has been done (Menberu et al., 2016; Günther et al., 2020). In drained peatland forests, this typically means at least
reducing drainage by blocking ditches, as well as removing or reducing tree cover, which should lead to a significant rise in
the WTL and a gradual return of wetland vegetation (Tarvainen et al., 2013; Maanavilja et al., 2015; Menberu et al., 2016).

Besides changes in the management of drained peatlands, the rapidly changing climate will also influence future $CH_4$ fluxes.
In Finland, mean annual temperatures are rising twice at the rate of global averages (Mikkonen et al., 2015). Precipitation, but
also evapotranspiration, are also expected to increase, causing changes in hydrological conditions as well (Ruosteenoja and
Jylhä, 2021). Previous studies have shown varying responses of peatland $CH_4$ fluxes to warming, largely depending on the
simultaneous changes in local WTL (Turetsky et al., 2008; Laine et al., 2019; Peltoniemi et al., 2016). However, the combined
effects of peatland management practices and climate change on $CH_4$ fluxes are insufficiently understood. To quantify green-
house gas fluxes from drained peatlands, Lehtonen et al. (2023) for example highlighted the need for a suitable mechanistic
model which we have utilized here.

In this study, we have simulated what would happen to $CH_4$ fluxes in a forestry-drained, nutrient-rich peatland during the
21st century if in 2020 there was a change in the management practice to 1) rotational forestry, 2) continuous cover forestry or
3) restoration to a wetland. The simulations were run throughout Finland under two climate scenarios, RCP 2.6 and RCP 4.5.
We have used the land surface model JSBACH that is driven by daily climate model data and simulates hydrology, vegetation
dynamics and carbon balance as well as the response of the ecosystem to climatic changes, extreme events and management
options. We coupled JSBACH with the YASSO soil carbon model and HIMMELI methane model to simulate relevant processes
in peatland carbon cycling. In this study, we investigated the simulated $CH_4$ flux as well as soil WTL, temperature and soil
carbon pools to understand 1) the combined effects of peatland management practices and climate change on future $CH_4$ fluxes
and 2) how $CH_4$ fluxes vary across Finland.



## 2 Model description and methodology

### 2.1 Land surface model JSBACH


JSBACH (Reick et al., 2013) is the land surface model of the Max Planck Institute for Meteorology Earth System Model (MPI-ESM) (Giorgetta et al., 2013) that simulates terrestrial energy, hydrology and carbon fluxes. Sub-grid scale heterogeneity is described through different vegetation types, in JSBACH called plant functional types (PFT) which are represented in the model through separate tiles within each grid cell. These are linked with a set of properties, such as phenology type or albedo,

that relate the PFTs to the processes accounted for in JSBACH (Reick et al., 2013). A detailed description of the whole model can be found in Reick et al. (2021). In this work we made site simulations with only PFT per site, assigning extra-tropical evergreen PFT for forests and peatland vegetation for restored and pristine peatland runs. We used a version of JSBACH3 that has been connected with Yasso soil carbon model ((JSBACH-peat; Goll et al., 2015) and which we applied to peatlands with WTL dynamics following Kleinen et al. (2020). Additionally, we coupled JSBACH-peat with HIMMELI methane model to

simulate peatland methane dynamics (Raivonen et al., 2017). Then, to simulate forest management on peatlands, we coupled JSBACH-peat with another model version of JSBACH which can account for forest growth within the forest PFTs in a similar way to Nabel et al. (2020) and is hereafter referred to as JSBACH-FOM. Below we discuss the model parts that have been modified and are most relevant for this study (Figure 1 a).

### 2.1.1 Soil water-table level control

WTL is very important in controlling peatland carbon dynamics in both drained and pristine peatlands. Li et al. (in preparation) implemented WTL in the peatland-YASSO in a drained peatland to partition the peat soil into anaerobic and aerobic fractions. We further coupled the peatland forest evapotranspiration with WTL and allowed WTL movement across the total peat column, which is important in drained peatlands. WTL was estimated from peat column water volume that is controlled by liquid precipitation, snow melt, evapotranspiration (ET) and run-off. The formulation of WTL used for pristine wetland was

made following the approach in Wania et al. (2009) and the potential evapotranspiration was used to estimate ET. In our implementation for drained peatlands the actual simulated forest ET was used to drive the water balance. This change was made to account for the impact of forest growth on the ET. The range of WTL was increased to 0.95 m to allow WTL to fluctuate in a deeper layer than the 0.3 m of the pristine wetland set-up. Moreover, the minimum fractional water content was increased to 0.65 from its default value of 0.25 implying that in our formulation it does not represent the original physical definition of

water holding capacity of spaghnum peat, but has to rather be considered a tuning parameter for adjusting the WTL variability to its observed level and range of variability.

### 2.1.2 Yasso soil carbon model

To simulate soil organic carbon and its decomposition, we used the Yasso soil carbon model that has been coupled with JS-BACH (Goll et al., 2015). Yasso divides soil carbon into slowly and rapidly decomposing pools consisting of carbon originating



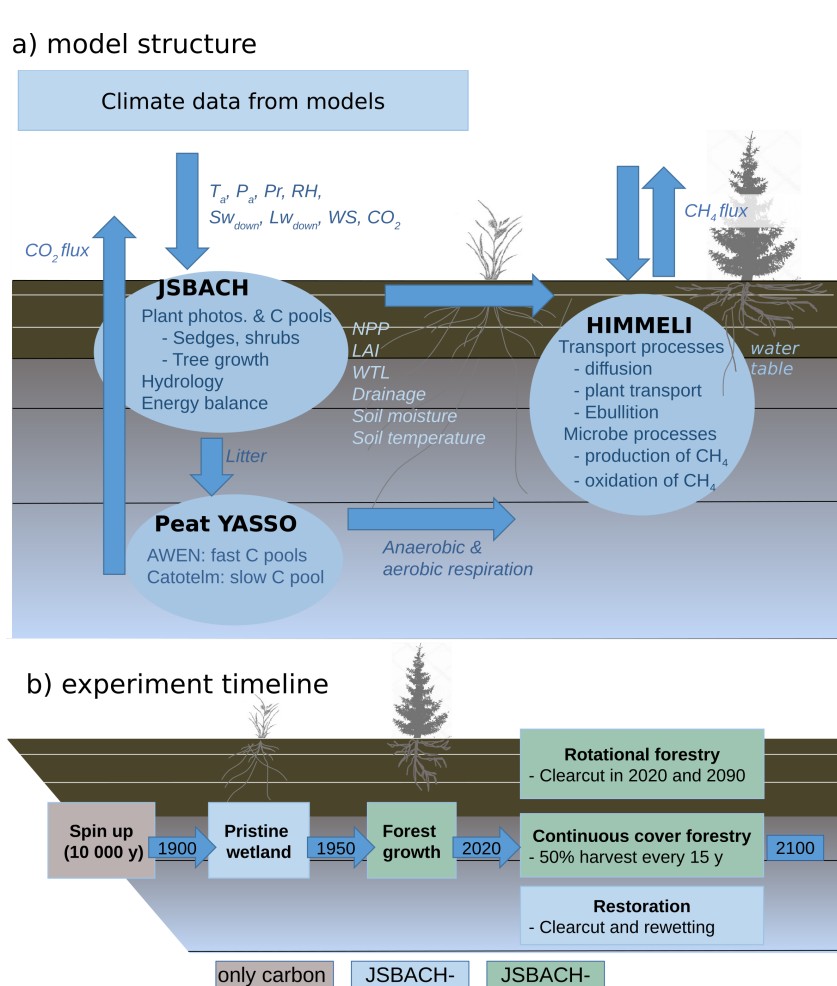

**Figure 1.** Schematics of a) how the models JSBACH, YASSO and HIMMELI are linked to each other and b) the experiment timeline in each model run. The acronyms in the figure are: $T_a$ = air temperature, $P_a$ air pressure, Pr = precipitation, RH = relative humidity, SW and LW = short-wave and long-wave radiation, WS = wind speed, NPP = Net Primary Production, LAI = Leaf Area Index, WTL = water-table level, C pools = carbon pools.

from either woody or non-woody plant structural parts. The four rapidly decomposing pools (i.e. AWEN; acid-soluble, water-soluble, ethanol-soluble and non-soluble) take into account the chemical composition of the litter input which in turn depends on the PFTs. Each pool consists of an above-ground and a below-ground part and in addition, there is the slowly decaying pool, called catotelm pool. The model parameters have been determined based on litter decomposition measurements worldwide in order to obtain realistic heterotrophic respiration rates for a range of conditions. In JSBACH-FOM, the model parameters

follow the calibration done in forest soils by Tuomi et al. (2009; Table A1). In JSBACH-peat, parameters follow the parametri-



sation used by Li et al. (in preparation; Table A2). A more detailed description of the model and parameter optimization can be found in Tuomi et al. (2009).

In the peatland implementation, the WTL further divides soil into oxic and anoxic layers. The decomposition of the anoxic fraction is slowed down from the oxic reference decomposition (Tables A1 and A2). The slowly decaying pool is the bottom
layer and the fast-decaying layer on top of that consists of below-ground and above-ground AWEN pools, in this order. The thickness of the anoxic and oxic layers is determined by the carbon contents of the pools and their bulk densities. The oxic and anoxic fractions of each carbon pool are determined by WTL depth in soil.

### 2.1.3  JSBACH-FOM

To simulate forest growth and harvest, we used JSBACH-FOM. It accounts for the age of the forest PFT and has a different
control of maximum forest leaf area index (LAI) compared to JSBACH-peat where maximum LAI is only dependant on the plant functional type. In JSBACH-FOM, it is dependent on available leaf biomass which is used to simulate forest regrowth (see Nabel et al., 2020; Wey et al., 2022). Change in total vegetation carbon is used as a basis for calculation of the growing forest. The number of trees per area, or stem number, is calculated from the total vegetation carbon assuming that the forests are in a self-thinning state. The biomass per individual tree can then be used together with allometric relationships to derive
the maximum LAI of the forest. The implementation of the forest harvesting and forest growth based on a maximum LAI is described in detail by Nabel et al. (2020). The setup of the model parameters for our simulations is explained in detail in appendix B. The harvest is done for each tile when forest age reaches the preset rotation time. Then, the forest is clearcut at the beginning of a year, and the age and size start from zero. The forest stand carbon pools are redistributed due to harvesting. The harvested carbon taking 77% of the above ground woody pool, is removed from the calculations. A half of the green carbon,
both above and below ground vegetation parts, go into the Yasso AWEN litter pools according to predefined fractions. The other half is located directly to the catotelm pool. Similarly, half of the below ground woody carbon, accounting for 30% of the growing stand total woody carbon, is distributed into the below ground litter pools, while the remaining half goes to the catotelm pool. In addition, the remaining 23% of the above ground woody carbon, accounting for the above ground growing stand, is distributed into below-ground litter pools (50%) and catotelm pool (50%). The redistribution of the cut forest stand
carbon to the soil carbon pools in the context of selection harvests is done in a similar way.

### 2.1.4  HIMMELI methane model

To simulate $CH_4$ fluxes in peat soils, we used the HelsinkI Model of MEthane buiLd-up and emIssion for peatlands (HIMMELI) which has been developed to simulate the build-up, transport and oxidation of $CH_4$ in peat soils (Raivonen et al., 2017). It is driven by soil temperature, WTL, LAI of aerenchymatous plants and the rate of anaerobic soil respiration. HIMMELI simulates
several microbial processes and transport pathways of $CH_4$, $CO_2$ and $O_2$, mainly $CH_4$ production and oxidation, aerobic respiration, ebullition, gas diffusion within the peat layer and transport in the plant aerenchyma. Unlike in the HIMMELI version used in JSBACH-PEAT, in the JSBACH-FOM, gas transport within plant aerenchyma was not included as there are



very little vascular plants with aerenchyma at drained peatlands (Laiho et al., 2003; Päivänen and Hånell, 2012). The methane model parameters used with JSBACH-FOM and pristine land simulations are given in Table A3.

We used a HIMMELI version that was modified by Li et al. (in preparation) to better suit for simulating also CH4 uptake, which is relevant at drained peatland. The modified version differs from the original in how the concentrations of compounds in soil layers are treated in the case of lowering WT.

## 2.2 Experiment design

The two setups of JSBACH used in this study differ in terms of biomass dynamics and soil carbon model parameterisation.
In JSBACH-FOM, the PFT was extra-tropical evergreen and in JSBACH-peat it was peatland vegetation. Peatland parameters from Hagemann and Stacke (2015) were used to describe the soil properties, e.g. soil porosity, saturated hydraulic conductivity, field capacity and wilting points and saturated moisture potential. Maximum root depth was set to 1.6 m because the model top soil layers tend to occasionally become relatively dry and thus unrealistically limiting photosynthesis.

The simulations were run forcing JSBACH with the regional EURO-CORDEX daily resolution climate data from the center
point of each mainland region in Finland (Jacob et al., 2014). The regional CORDEX models are forced by coarse resolution global CMIP6 climate models. The Euro-CORDEX models are validated and bias-corrected by Finnish observations (Räisänen and Räty, 2013; Räty et al., 2014). Three climate models (CanESM2, MIROC5 and CNRM-CM5) based on two emission scenarios, RCPs 2.6 and 4.5, were used in order to better understand climate-related uncertainty in the results. For RCP 2.6, we only used models MIROC5 and CNRM-CM5 as not all required drivers were available in CanESM2.

The initial state of JSBACH can either be adopted from observations or produced in a spin-up run, where selected state variables are usually taken to an equilibrium state under given climate. For certain very slowly evolving state variables, such as carbon storages of pristine peatlands, a spin-up to equilibrium would require unrealistically long time and spin-up runs have to be interrupted prior to equilibrium. Thus, to account for soil carbon accumulation, we used a spin up of 10 000 years, the time period since the last ice age, running only the soil carbon processes of the model system using climate data from the years
1900-1930 together with NPP produced with respective climate and CO2 concentration of the year 1900. The resulting carbon stock from the spin up run was approximately 106-120 kg/m$^2$ and the peat depth 3.4-3.9 m which is within the range expected from previous studies that have estimated total carbon stocks of peatlands in Finland (Turunen et al., 2002; Juutinen et al., 2013).

To account for the impact of the transient change of the climate and the CO2 concentration on the system state, we continued
a full pristine wetland run from 1900 to 1950 with increasing CO2 concentration (Figure 1 b). In 1950, the model version was changed to JSBACH-FOM, conifer seedlings were planted, and the model was run until 2020 when the forest was 70 years old, with 640 trees per ha. JSBACH-FOM does not explicitly simulate ditches but model drainage and runoff do remove water from the soil. In 2020, each simulation was split into three management options which were continued until the end of the century. The first option was rotational forestry in which the forest was harvested every 70 years, i.e. in 2020 and 2090. The harvest
in this option was performed as a clearcut, where all the living stand was removed and the harvest residue was located to the YASSO pools.



In the second option, continuous cover forestry, we did a selection harvesting by removing 50 % of the woody biomass every 15 years (Juutinen et al., 2021). The harvest was simulated by using JSBACH-FOM without applying the clearcut -growth cycles controlled through FOM. Instead, the state of the stand carbon storages and the consequent changes in the soil carbon storages were modified in simulation restarts timed to take place in constant time intervals. In each restart a pre-determined fraction of the stand biomass was removed with identical re-location fractions of the biomass to the wood products and soil carbon pools as in the baseline clearcut case. Effectively, the manipulation returns the stands to an earlier growth phase at the stand growth curve.

The third option was restoration to an undrained peatland in which there was a clearcut in 2020 and the simulation was continued using JSBACH-peat until 2100.

## 2.3 Flux evaluation data

To validate the simulated methane fluxes, we used chamber measurements of $CH_4$ fluxes from 5 forestry-drained, 27 restored and 6 pristine peatland sites in Finland, established for various research projects. Measurement points had intact ground vegetation and tree roots; thus, the measurements include all components of $CH_4$ flux between forest floor and the atmosphere.

Fluxes were measured using portable greenhouse gas measurement devices (LI-7810, LI-COR; M-GGA-918, ABB; Gasmet DX4015, Gasmet), except for the oldest measurements for which gas samples were taken into syringes and analyzed in the laboratory of the Natural Resources Institute Finland with a gas chromatograph equipped with an FI detector for methane. For all measurements, a similar opaque round metal chamber (height 30 cm, diameter 31.5 cm) equipped with a fan for air mixing, was utilized. Measurements were carried out on a biweekly to monthly interval during the snow-free season (May–October). A single measurement lasted either 2–5 minutes (portable devices) or 25 minutes (manual sampling for laboratory analysis).

In addition to the $CH_4$ flux, WTL and soil temperature at 5 cm depth during the flux measurements were measured. WTL was measured from a well that was a perforated plastic tube installed into the soil when establishing the study sites. It was measured either manually or utilizing the Odyssey Capacitance Water Level Loggers (Dataflow Systems Ltd). Soil temperature was measured either utilizing manual temperature probes or with iButton DS1921G loggers (Maxim Integrated).

The five forestry-drained sites were located in southern to central Finland (4 sites) and northern Finland (1 site). Each site had a control treatment without cuttings and a partial harvest treatment (thinning, overstorey harvesting or strip cutting). Three of the sites had in addition a clearcut treatment. The flux measurements ranged from the 1st to the 9th year after cuttings during the years 2016–2021, depending on the site. The experiment setup and thus also the number of measurement points varied from site to site; yet there were always several points per treatment covering the typical soil moisture variations from the vicinity of the nearest ditch to the mid-strip. All the forestry-drained sites had been drained for several decades before the measurements. They were originally drained for practical forestry purposes with a typical ditch spacing of 40 meters and a ditch depth of ca. 1 meter. The sites represented a wide range of site types from oligotrophic to eutrophic peatlands.

The restored and pristine sites located in southern and central Finland. The restored sites were rewetted, either by damming or by filling up the ditches between 1993 and 2020. The sites had previously been drained for forestry: thus, they had a similar ditch spacing and ditch depth as the drained sites in this study. During restoration, ditch banks were cleared from trees, but no





other tree stand management was done. CH$_4$ flux measurements were done during 2021 from several points per site. At the restored sites, measurement points were located on strips and formerly filled or dammed ditches. The sites again represented a wide range of site types from oligotrophic to eutrophic.

## 2.4 Data analysis

In addition to the simulated CH$_4$ fluxes, we investigated management impacts on peatland WTL, carbon pools in the upper (fastly-decaying) peat layers, LAI and soil temperature in order to understand the controls of CH$_4$.

To estimate trends in CH$_4$ fluxes and the environmental variables controlling them between 2020–2100, we used Mann-Kendall trend test which is used to determine whether there is a monotonic upward or downward trend in a time series (Mann, 1945; Kendall, 1948). The test was calculated for the continuous cover forestry and restoration options and for each climate

model and region separately. The trends were not calculated for rotational forestry, as the clearcutting in 2090 interrupts any linear trends in the timeseries. The magnitude of the trend was estimated using Sen's slope estimator which is used to quantify significant linear trends in a timeseries Sen (1968). Both were calculated using the "trend" package, version 1.1.5. in R (Pohlert, 2023).

## 3 Results

## 3.1 Model validation


In both forestry-drained and restored peatlands, the simulated CH$_4$ fluxes matched measured fluxes reasonably well (Figure 2). In restored peatlands, both models and measurements showed emissions to increase with increasing soil temperatures (Figure 2 a). In forested peatlands, measurements and model results showed a weak sink of CH$_4$ when WTL was low (below -15 cm) and that the sink weakened with higher WTL (Figure 2 b). The measurements showed occasional emissions regardless of WTL

while the model results showed emissions only in the clearcut option when WTL was high. The CH$_4$ sinks were strongest in the mature forest (control sites) in both simulations and measurements.

## 3.2 Environmental controls

Clearcutting and restoration had the largest impacts on WTL and fastly-decaying carbon pools but had no impact on summer soil temperatures (Figure 3). Summer WTL rose on average by 30 cm right after the clearcut and by 20 cm after restoration,

while there was only a slight rise in WTL after continuous cover forestry (Figure 3). After the clearcut, WTL remained higher than in the continuous cover forestry option for 20–30 years and then decreased before the next clearcut in 2090. The higher WTL after a clearcut compared to restoration was due to differences in the LAI — in JSBACH-PEAT, LAI recovered rapidly after the restoration while after the clearcut, forest LAI took considerably longer to grow, keeping transpiration rates lower during the first years. Fastly-decaying pools increased following all management practices due to harvest residue but started

do decrease due to the decomposition of the harvest residue in the rotational forestry and restoration options.





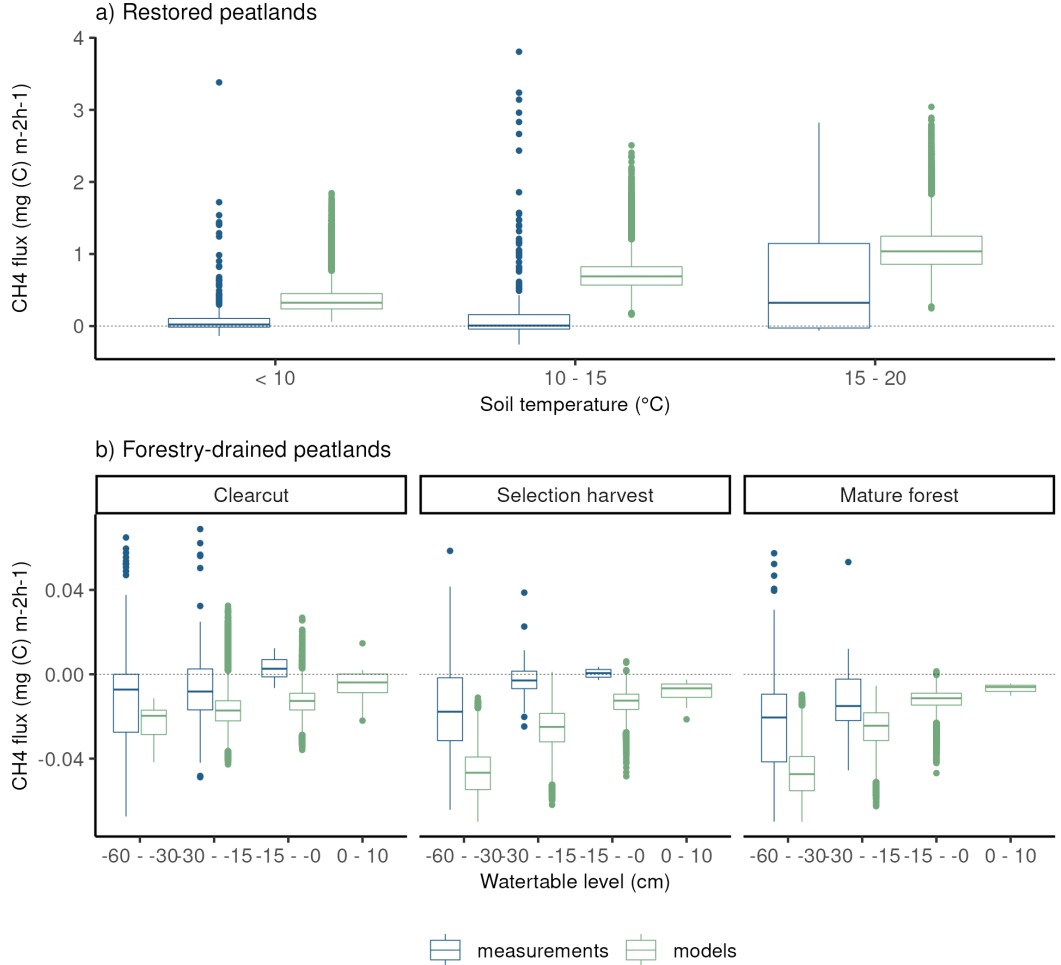

**Figure 2.** Comparisons between modelled and measured daily CH$_4$ fluxes. Panel a) shows CH$_4$ fluxes in soil temperature classes in restored peatlands and panel b) shows CH$_4$ fluxes in watertable level classes in forestry-drained peatlands.

Towards the end of the century under RCP 4.5, there was a slight positive trend in WTL in the continuous cover forestry option, particularly in northern and central Finland although the trend was not significant in all models and the estimated Sen's slopes varied considerably between models (Table A5). In the restored option, Uusimaa and other southern regions had a slight positive trend in WTL as well but western coastal regions had a slight negative trend that was also not significant in all

models (Table A4). Fast-decaying pools had a clear negative (positive) trend throughout Finland in the restoration (continuous cover forestry) option but the trend was stronger (weaker) in southern and central Finland. Summer soil temperatures increased throughout the country with little regional variation in both management options.



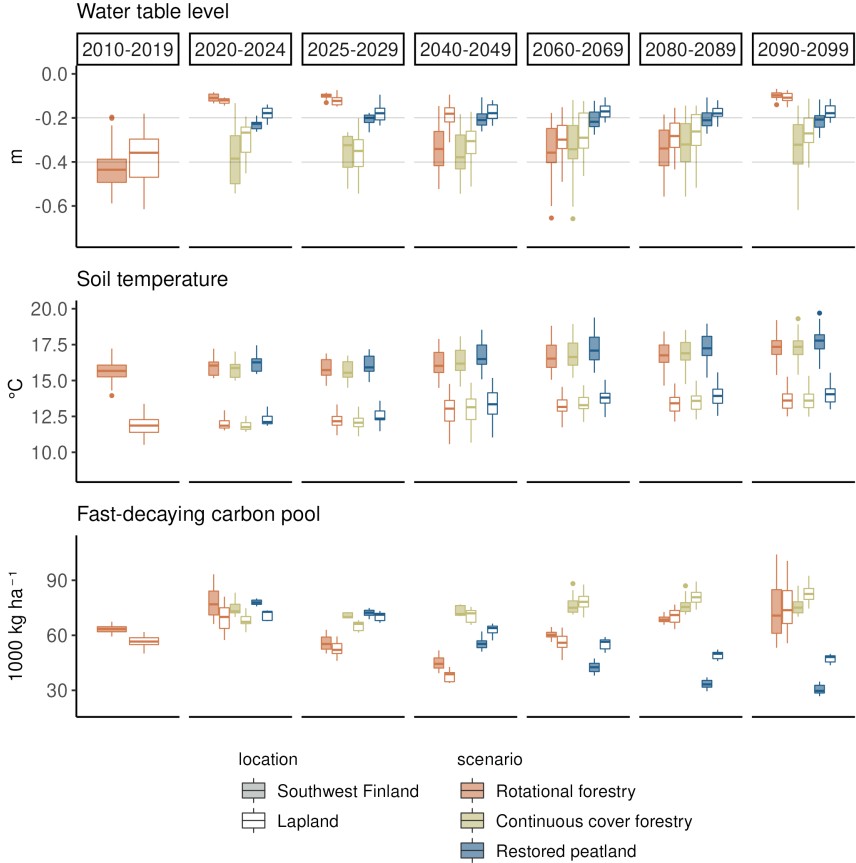

**Figure 3.** The impact of forest management on summer WTL, soil temperature (-22 cm below ground) and fast-decaying carbon pools under RCP scenario 4.5. Each boxplot shows variation created by yearly variation and differences in the three climate models. Two regions, Southwest Finland and Lapland, are presented to show regional variability across Finland. The first column (2010–2019) shows the situation in a mature forest before the beginning of the different management scenarios. The next two columns (2020-2024 and 2025-2029) show the situation during the first decade of the different management scenarios and the last three columns (2040-2049, 2060-2069, 2080-2089 and 2090-2099) show the development of the variables during the later decades.

## 3.3 The impact of management practices and climate on CH$_4$ fluxes

### 3.3.1 Restoration

Before the first harvest in 2020, the mature forest was a small sink of CH$_4$ with an average annual sink of 2 kg (C) ha$^{-1}$ in southwestern Finland and 1.8 kg (C) ha$^{-1}$ in Lapland (Figure 4). Following the first harvest in 2020, the restoration option turned to a strong source of CH$_4$ (20-45 kg (C) ha$^{-1}$a$^{-1}$). The emissions in southwestern Finland were on average twice as



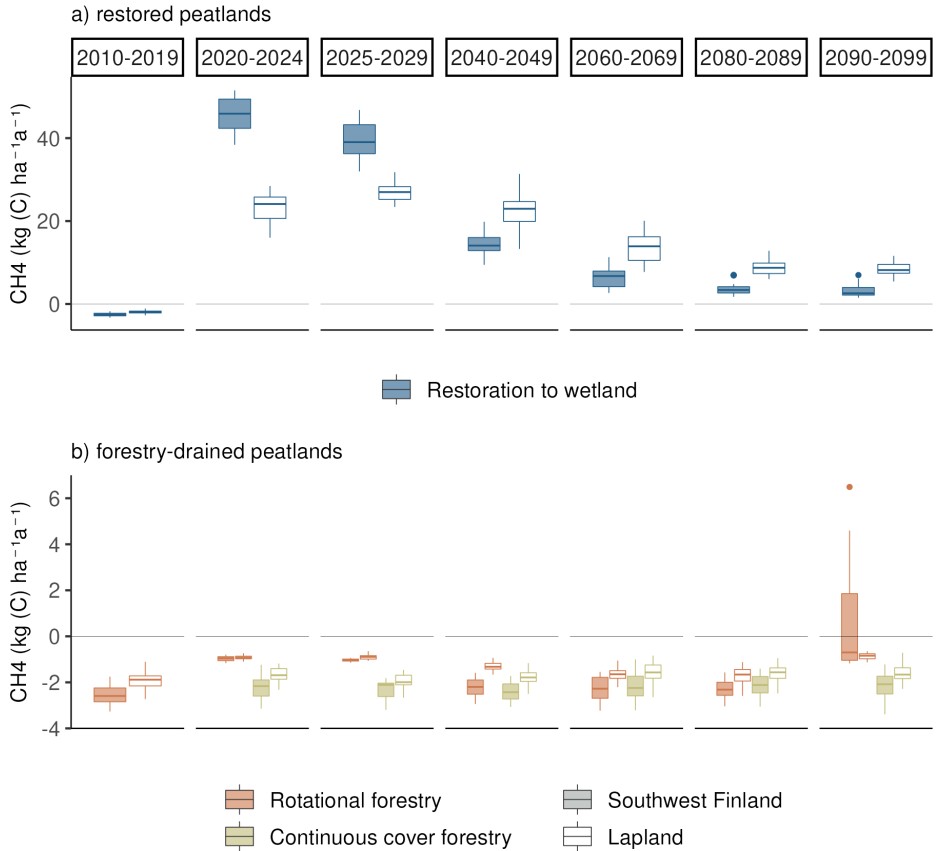

**Figure 4.** Average annual CH4 flux in (a) restored peatlands and (b) under forest management scenarios (rotational and continuous cover forestry) during the 21st century under RCP scenario 4.5 in Southwestern Finland and Lapland. Each boxplot shows variation in the $CH_4$ flux created by yearly variation and differences in the three climate models. The first column (2010-2019) shows the situation in a mature forest before the beginning of the different management scenarios. The next two columns (2020-2024 and 2025-2029) show the situation during the first decade of the different management scenarios and the last three columns (2040-2049, 2060-2069, 2080-2089 and 2090-2099) show the development of CH4 flux during the later decades.

large as in Lapland during the first decade after restoration. After the first decade, emissions in southern Finland decreased to the same level as emissions in Lapland.

During the 21st century, there was a strong negative trend in $CH_4$ emissions, leading to smaller emissions by 2100 (Figure 5 and Table A4). The slope of the trend was largest in eastern Finland and the smallest in Lapland, causing Lapland to have higher emissions in later decades compared to other regions (Figure 4).





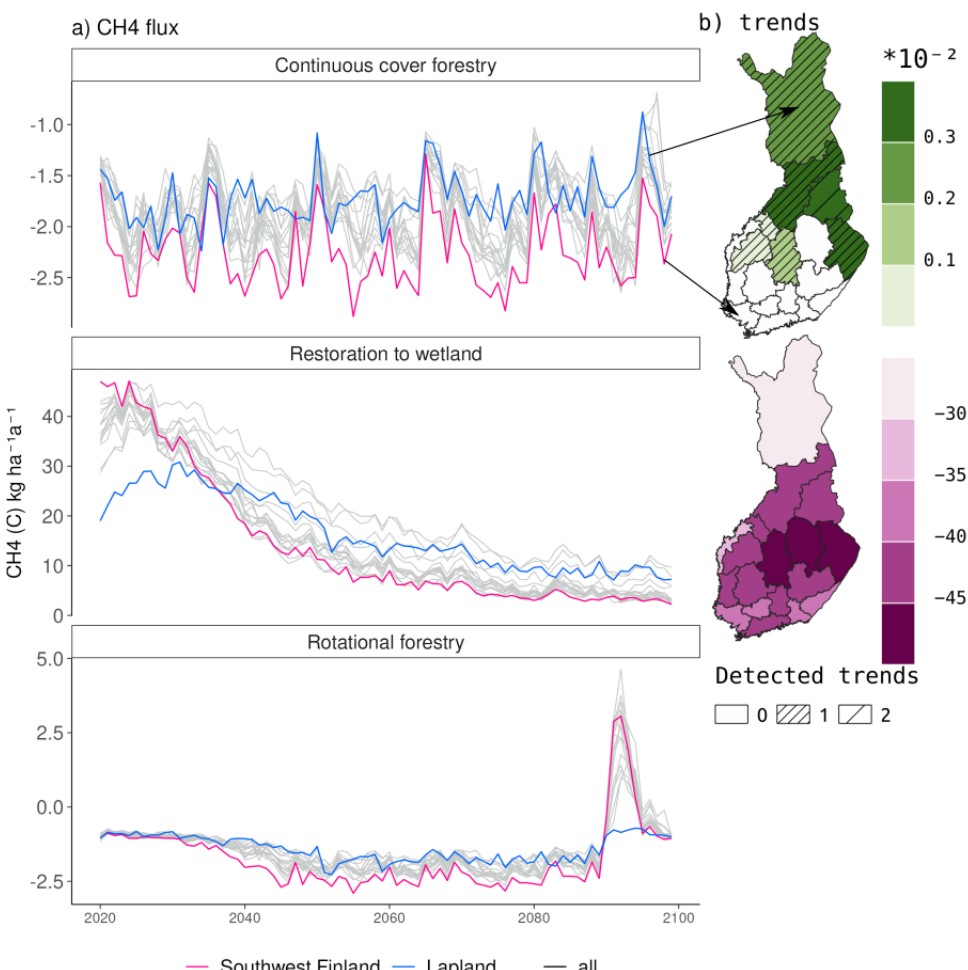

**Figure 5.** Regional variation in the CH$_4$ flux over Finland under RCP 4.5. Panel a) shows the mean annual flux averaged over three climate models in the three management options. Panel b) shows the average Sen's slope describing a linear trend in a timeseries in restored peatlands and in the continuous cover forestry management option. The dashed lines indicate how many of the three climate models estimated a statistically significant (p < 0.05) trend. No trend was calculated for rotational forestry as the clearcutting in 2090 would disrupt any linearity in the trend.

### 3.3.2 Continuous cover forestry

In continuous cover forestry, the start of harvesting in 2020 had very little impact on the CH$_4$ sink which remained approx-
imately at -1– -2 kg (C) ha$^{-1}$a$^{-1}$ (Figure 4). The sink was stronger in southern Finland. In later decades, there was a slight positive trend in the CH$_4$ in the northern and eastern parts of Finland under RCP 4.5, indicating a weakening CH$_4$ sink (Figure 5). However, there was also considerable variability between the climate models (Table A5).





### 3.3.3 Rotational forestry

Rotational forestry had a stronger impact on the $CH_4$ fluxes compared to continuous cover forestry, with the average sink

weakening to 1 kgha$^{-1}$a$^{-1}$ in both southwestern Finland and Lapland under RCP scenario 4.5 (Figure 4). In the first decades after clearcutting, there was little regional variation in the $CH_4$ sink but after 2030, the sink decreased in southern Finland while there was little change in Lapland throughout the century (Figure 5 a).

After the clearcut in 2020, the soil was on average a $CH_4$ sink despite occasional emissions during summer (Fig. 6). However, after the clearcut in 2090, the sporadic emissions increased, leading the soil to turn to a source of methane for several years

under RCP 4.5 in most parts of the country(Figure 4). Number of days with emissions during summer increased the most in southern and central parts of the country where over third of the days during summer months had emissions, compared to less than 10 after the 2020 harvest. In Lapland, the increase in emission days was smallest and on average, the soil stayed a $CH_4$ sink. Soil temperatures were on average 1-2 degrees warmer in 2090 than in 2020 while WTL was only slightly higher, by 1-2 cm. Anoxic conditions in the fast-decaying upper carbon pools increased considerably after the 2090 clearcut, likely due to the

slightly higher WTL and increases in the fastly-decaying pool.

### 3.4 The impact of climate scenarios

In the forestry-drained peatlands, there were stronger regional differences under RCP 2.6 compared to RCP 4.5 with a stronger $CH_4$ sink in both management options in southern Finland compared to northern Finland (Figure A1). Similarly, regional differences in WTL were stronger under RCP 2.6 (Figure A2). In southern Finland, average WTL stayed below -30 cm throughout

the century while average WTL in northern Finland was approximately -20 cm. Thus, in both management options, the cumulative sinks calculated over 2020-2100 were stronger under RCP 2.6 in all counties except Lapland where the sink was weaker (Table 1). Under both management options, the cumulative sink was approximately 30-40 kg (C) ha$^{-1}$ stronger in the continuous cover forestry option compared to rotational forestry.

In the restoration option, $CH_4$ emissions during the first decades after restoration were lower under RCP 2.6, particularly in

Lapland where emissions were approximately half compared to RCP 4.5 (Figure A1). Linear trends were stronger in southern and western Finland under RCP 2.6 but weaker in eastern and northern Finland (Table A4). Thus, the cumulative emissions were slightly higher in southern and western Finland under RCP 2.6 and lower in eastern and northern Finland (Table 1).

## 4 Discussion

### 4.1 Management impacts on $CH_4$ fluxes

According to our simulations, $CH_4$ fluxes on drained boreal peatland forests vary depending on the management practices. Considering harvesting practices, clearcutting resulted in a considerable rise in WTL and a consequent weakening of the $CH_4$ sink with also occasional emissions. In previous empirical studies, the forest $CH_4$ sink has decreased following a clearcutting and some sites have turned to sources of $CH_4$ during the first years following a clearcutting (Wu et al., 2011; Korkiakoski et al.,



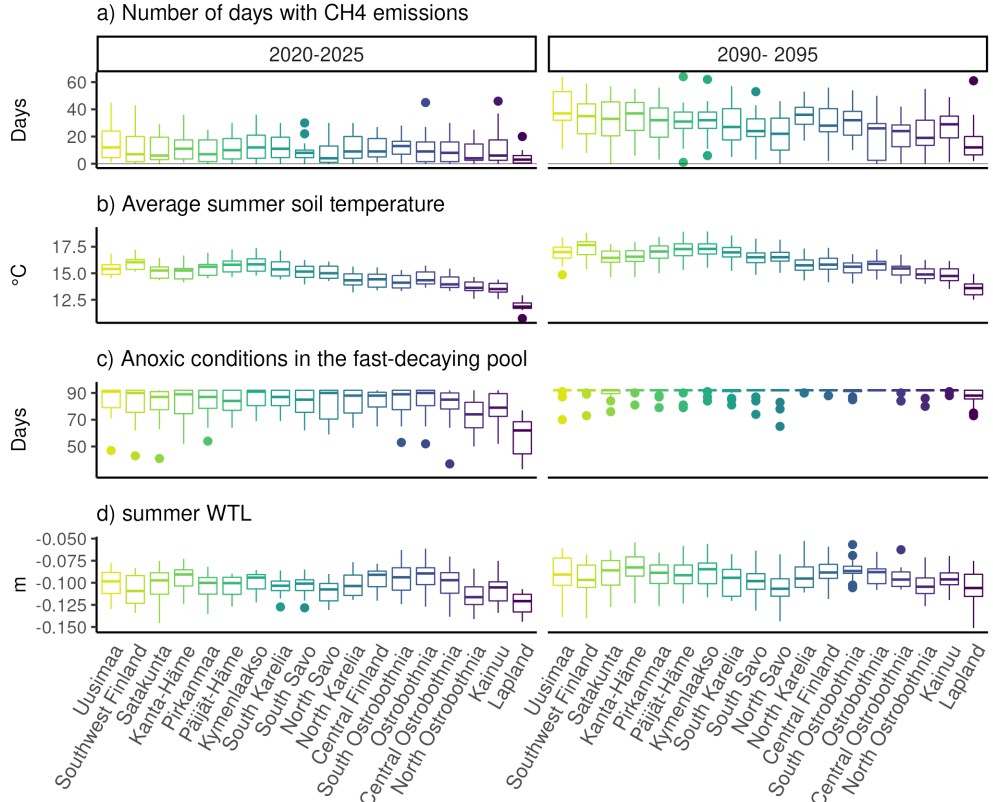

**Figure 6.** Differences in CH$_4$ emissions and their environmental controls following the clearcuts in 2020 and 2090 under RCP 4.5. Panel a) shows the number of days during a year with any $CH_4$ emissions, panel (b) shows mean temperatures during June, July and August, panel c) shows the number of days when any part of the fast-decaying soil carbon pools had anoxic conditions and panel d) shows mean summer WTL.

2019). However, some studies have also found no considerable changes in CH$_4$ fluxes following a clearcutting (Saari et al.,
2009). Our results indicate that the CH$_4$ sink may stay weakened following the clearcut for over a decade which is in line with
observations made by Wu et al. (2011) and highlight the considerable long-term effects of harvesting practices on peatlands.
In comparison, selection harvesting had only minor effects on WTL and CH$_4$ sink. While this is in line with previous studies
(Sundqvist et al., 2014; Korkiakoski et al., 2020), it should be noted that ditches and logging trails may still be sources of
CH$_4$, weakening the CH$_4$ uptake also after selection harvests (Korkiakoski et al., 2020). In both harvesting options, variation
in the CH$_4$ flux between southern and northern Finland was small and seemed to follow variations in WTL, indicating that
fluctuations in hydrological conditions due to climatic variability may control forest CH$_4$ fluxes (Ojanen et al., 2010).

Following restoration, the simulated peatlands turned to considerable sources of CH$_4$. This increase in CH$_4$ emissions has
been associated with higher WTL as well as recovery of peatland vegetation capable of CH$_4$ transport directly to the atmosphere



**Table 1.** Cumulative CH$_4$ fluxes between 2020 and 2100 averaged over the climate models.

| location | Rotational forestry | | Continuous cover forestry | | Restoration to wetland | |
|---|---|---|---|---|---|---|
| | RCP 4.5 | RCP 2.6 | RCP 4.5 | RCP 2.6 | RCP 4.5 | RCP 2.6 |
| Uusimaa | -118 | -172 | -161 | -208 | 1232 | 1383 |
| Southwest Finland | -140 | -171 | -183 | -207 | 1117 | 1247 |
| Satakunta | -117 | -158 | -157 | -190 | 1187 | 1197 |
| Kanta-Häme | -110 | -167 | -148 | -201 | 1191 | 1444 |
| Pirkanmaa | -117 | -161 | -155 | -192 | 1262 | 1267 |
| Päijät-Häme | -119 | -166 | -157 | -196 | 1218 | 1338 |
| Kymenlaakso | -122 | -170 | -161 | -203 | 1067 | 1405 |
| South Karelia | -127 | -177 | -167 | -205 | 1085 | 936 |
| South Savo | -125 | -159 | -159 | -193 | 1218 | 1136 |
| North Savo | -124 | -131 | -157 | -159 | 1369 | 1656 |
| North Karelia | -97 | -140 | -135 | -173 | 1823 | 1262 |
| Central Finland | -101 | -146 | -135 | -176 | 1561 | 1625 |
| South Ostrobothnia | -111 | -149 | -146 | -179 | 1222 | 1709 |
| Ostrobothnia | -141 | -145 | -175 | -175 | 865 | 1684 |
| Central Ostrobothnia | -119 | -146 | -150 | -177 | 1227 | 1522 |
| North Ostrobothnia | -113 | -117 | -143 | -143 | 1331 | 1780 |
| Kainuu | -98 | -111 | -129 | -137 | 1696 | 1520 |
| Lapland | -114 | -76 | -140 | -89 | 1316 | 1105 |

from the anoxic soil layers (Putkinen et al., 2018; Urbanová and Bárta, 2020). Contrary to the forested peatlands, there were

considerable differences between southern and northern Finland, particularly during the first decade after restoration when CH$_4$ emissions in southern Finland were nearly twice as large as in northern Finland. This difference was likely due to soil temperatures which were nearly 5 °C higher in southern Finland and which control particularly CH$_4$ production (van Hulzen et al., 1999; Turetsky et al., 2014).

### 4.2 Long-term trends and the impact of climate

In restored peatlands, the simulated CH$_4$ emissions decreased under both climate scenarios towards the end of the century. This trend was likely due to the decreasing fast-decaying carbon pool in which a large majority of the CH$_4$ emissions are produced in the model. This pool was high immediately after restoration due to harvest residue and started then decreasing with the decomposition of the residue. This decrease in both the fast-decaying carbon pools and CH$_4$ emissions was particularly strong in southern Finland where decomposition of organic material was higher due to higher temperatures. Furthermore, under RCP

2.6, decomposition of the fast-decaying pools was slower due to lower temperatures throughout the century, causing the net emissions to be slightly higher in southern and western parts of the country despite lower temperatures typically decreasing





CH$_4$ emissions (Lai, 2009). Previous research suggests that particularly nutrient-rich peatlands may have strong CH$_4$ emissions immediately after restoration, followed by a decrease over time (Wilson et al., 2016). However, the decay of fast-decaying pools is possibly overestimated in the simulations which may lead to an underestimation of CH$_4$ emissions in the latter part of the

century. Still, long-term monitoring studies following restoration of drained peatland forests are still scarce, the majority of the studies having been done in cutover peat extraction areas (e.g. Wilson et al. 2016). Peatland plant species also differ in their CH$_4$ transport rate and efficiency as well as their contribution to CH$_4$ formation through substrate production (Dorodnikov et al., 2011; Ge et al., 2023). Consequently, gradual changes in vegetation composition following restoration may have significant impacts on the ecosystem CH$_4$ flux although this was not considered in our simulations. Nonetheless, the impact of warming

on CH$_4$ fluxes has been conflicting in previous studies and is strongly dependent on peatland hydrology and its development (Turetsky et al., 2008; Peltoniemi et al., 2016). Our results thus highlight the need for long-term research on restored peatland forests in order to better understand the impacts of restoration and warming.

In the continuous cover forestry option, variations in the CH$_4$ sinks seemed to follow trends in WTL, with a weakening sink when WTL increased and a strengthening sink in areas where WTL decreased. This was particularly noticeable under the RCP

2.6 scenario where regional differences in WTL, and consequently in CH$_4$ fluxes, were stronger. However, it is important to notice that the trends in WTL were quite uncertain and depended largely on the climate model, which can be due to uncertainties in future precipitation patterns (Ruosteenoja and Jylhä, 2021). Previous study by Gong et al. (2012) found WTL to decrease towards the end of the century in drained peatlands but also that changes in drained peatlands were mostly smaller than in pristine peatlands. Understanding the changes in precipitation and their further impact on peatland hydrology is essential to

accurately project future GHG fluxes and their climatic impacts.

In the rotational forestry option, the CH$_4$ sink started to strengthen a decade after the clearcut in most parts of Finland aside from Lapland, which is likely due to a lowering of the WTL following forest regrowth and increased evapotranspiration. In 2090, the increased emissions following the second clearcutting were likely due to temperature increases together with increased anoxic conditions in the upper peat soil. The increased anoxic conditions resulted from a slight increase in WTL and

fastly-decaying carbon pools, resulting in a thicker anoxic layer compared to an oxic layer. However, it should also be noted that the variation in CH$_4$ fluxes following the clearcutting was considerably larger than in the previous decades. Under RCP 2.6, rotational forestry peatlands stayed as sinks despite similar increases in WTL, possibly due to lower soil temperatures.

### 4.3    Methodological limitations

The recovery of vegetation after forest management practices has been kept simplified here and may affect some of the results.

In the continuous cover forestry option, the harvest removes 50 % of the total stand biomass of the forest rather than removing specific trees. In practice this means that the harvesting decreases the total green and woody biomass of the forest but the forest growth resembles that of a mature forest rather than a mixture of young and old forest which then leads to a rapid recovery of the forest after harvesting. This means that the impacts of selection harvesting on WTL and consequently CH$_4$ fluxes may have been underestimated. In the rotational forestry option, the recovery of vegetation is slow after the harvest as the simulation

of pioneer species and natural tree/shrub regeneration are not included in JSBACH. This leads to a prolonged period of very





high WTL but might also underestimate the CH$_4$ transport through peatland species such as *Eriophorum vaginatum* which might be considerable following a clearcutting (Hamberg et al., 2019). Furthermore, the simulation did not include thinnings that are commonly done in Finland also in rotational forestry. These would likely have mediated the drop in WTL between the clearcuts and increased harvest residue in the soil, possibly slowing down the strengthening of the CH$_4$ sink particularly in

southern and central Finland in between clearcuts. In restored peatlands, the recovery of pristine vegetation may take a longer time after restoration compared to the very rapid recovery simulated by JSBACH.

We ran the simulations for each Finnish county to study the effect of climatic variation while keeping the required computational power at a reasonable level. While most geographic and climatic variation were adequately represented by this approach, northernmost Finland was largely represented by one single county, which means that some of the variation in northern boreal

regions is likely missing. However, there are very few forestry-drained peatlands in the northernmost regions in Finland, particularly in northernmost Lapland, and official reporting such as the national forest inventory is often published in county level (Natural Resources Institute Finland, 2023).

The functioning of peatland ecosystems and their methane fluxes can vary considerably depending on local environmental and climatic conditions such as microtopography, nutrients, hydrological conditions and past use (e.g. Lai, 2009). Thus,

country-wide simulations of hypothetical nutrient-rich peatlands with deep peat layers are unlikely to represent realistic conditions on any single site. Rather, this approach allows us to estimate and compare possible impacts of peatland management and climate and their feedbacks throughout a wide geographic range.

## 5    Conclusions

We simulated the impact of management practices and climate on CH$_4$ fluxes from forestry-drained boreal peatlands using the

land surface model JSBACH. Our simulations showed that restoration turned peatlands to sources of CH$_4$ but the magnitude of emissions varied regionally with larger emissions in southern Finland than in northern Finland. Furthermore, emissions decreased towards the end of the century as harvest residue diminished from the upper peat layers. In forested peatlands, clearcutting had a stronger weakening effect on the forest CH$_4$ sink compared to selection harvesting and the effect was stronger towards the end of the century under RCP 4.5. Water-table level was found to have a strong control on the CH$_4$ fluxes,

particularly on forested peatlands.

*Code and data availability.* The JSBACH model can be obtained from the Max Planck Institute for Meteorology, where it is available for the scientific community under the MPI-M Sofware License Agreement (https://mpimet.mpg.de/en/research/modeling, last access: 19 December 2023). Data used in this study is available by request from the authors.



## Appendix A

**Table A1.** Yasso parameters for JSBACH-FOM. Decomposition parameters are given for above-ground (ag) and below-ground (bg) AWEN (acid-hydrolyzable, water-soluble, ethanol-soluble and neither hydrolyzable or soluble) carbon pools. The parameters are Reference decomposition rate, anoxic decomposition modifier, peat decomposition modifier, anoxic peat decomposition factor, and anoxic and oxic decomposition rate. The proportion of soil carbon from N to A pool and the relationship between AWEN to H (humus) and N to H is also given.

| Layer / Factor | Ref. Dec. Rate | Anoxic Dec. Mod. | Peat Dec. Mod | Anox. Peat | Dec. Rate Anox | Dec. Rate Oxic |
|---|---|---|---|---|---|---|
| ag A | 0.72 | 0.35 | 1 | 0.35 | 0.252 | 0.72 |
| ag W | 5.9 | 0.35 | 1 | 0.35 | 2.065 | 5.9 |
| ag E | 0.28 | 0.35 | 1 | 0.35 | 0.098 | 0.28 |
| ag N | 0.031 | 0.35 | 1 | 0.35 | 0.01085 | 0.031 |
| bg A | 0.72 | 0.25 | 1 | 0.25 | 0.18 | 0.72 |
| bg W | 5.9 | 0.25 | 1 | 0.25 | 1.475 | 5.9 |
| bg E | 0.28 | 0.25 | 1 | 0.25 | 0.07 | 0.28 |
| bg N | 0.031 | 0.25 | 1 | 0.25 | 0.00775 | 0.031 |
| H | 0.0064 | 0.04 | 0.5 | 0.02 | 0.000128 | 0.0032 |
| N to A | 0.83 | | | | | |
| N to W | 0.01 | | | | | |
| N to E | 0.02 | | | | | |
| AWEN to H/ N to H | 0.0045 | | | | | |



**Table A2.** Yasso parameters for peatland version of JSBACH. Decomposition parameters are given for above-ground (ag) and below-ground (bg) AWEN (acid hydrolyzable, water soluble, ethanol soluble and neither hydrolyzable or soluble) carbon pools. The proportion of soil carbon from N to A pool and the relationship between AWEN to H (humus) and N to H is also given.

| Layer / Factor | Ref. Dec. Rate | Anoxic Dec. Mod | Peat. Dec. Mod | Anoxic Peat | Dec. Rate Anoxic | Dec. Rate Oxic |
|---|---|---|---|---|---|---|
| ag A | 0.72 | 0.2 | 1 | 0.2 | 0.144 | 0.72 |
| ag W | 5.9 | 0.2 | 1 | 0.2 | 1.18 | 5.9 |
| ag E | 0.28 | 0.2 | 1 | 0.2 | 0.056 | 0.28 |
| ag N | 0.031 | 0.2 | 1 | 0.2 | 0.0062 | 0.031 |
| bg A | 0.72 | 0.2 | 1 | 0.2 | 0.144 | 0.72 |
| bg W | 5.9 | 0.2 | 1 | 0.2 | 1.18 | 5.9 |
| bg E | 0.28 | 0.2 | 1 | 0.2 | 0.056 | 0.28 |
| bg N | 0.031 | 0.2 | 1 | 0.2 | 0.0062 | 0.031 |
| H | 0.0016 | 0.2 | 0.125 | 0.024 | 4E-05 | 0.0002 |
| N to A | 0.83 | | | | | |
| N to W | 0.01 | | | | | |
| N to E | 0.02 | | | | | |
| AWEN to H/ N to H | 0.0045 | | | | | |



**Table A3.** HIMMELI model parameters in JSBACH-FOM and JSBACH-PEAT.

| | unit | JSBACH-FOM | JSBACH-peat |
|---|---|---|---|
| Half-life of supersaturated $CH_4$ | | 30*1800 | 30*1800 |
| Supersaturation requirement for ebullition | | 1.04 | 1.0 |
| Potential rate of aerobic respiration at 10 °C | mol m$^{-3}$s$^{-1}$ | 3e-6 | 1e-5 |
| Potential oxidation rate at 10 °C | mol m$^{-3}$s$^{-1}$ | 1.5e-5 | 1e-5 |
| Michaelis constant for $CH_4$ in oxidation | mol m$^{-3}$ | 0.015 | 0.03 |
| Fraction of anaerobic respiration becoming $CH_4$ | – | 0.03 | 0.17 |
| Specific leaf area of gas-transporting plants | m$^2$ kg$^{-1}$ | 150000.0 | 15.0 |




**Table A4.** Linear trends and their uncertainty related to climate models of CH$_4$ fluxes, soil respiration, summer WTL and summer soil temperature in the Finnish counties in restored peatlands. The values show Sen's slope parameter averaged over the climate models and their variability. Bolded values indicate significant trends (p <0.05) as estimated by the Mann-Kendall trend test. [1] indicates only one model showed a significant trend, [2] indicates that two models showed a significant trend.

| location | CH$_4$ (*10$^{-2}$ kg ha$^{-1}$ a$^{-1}$) RCP 4.5 | | | RCP 2.6 | | | Fast decaying pools (kg ha$^{-1}$ a$^{-1}$) RCP 4.5 | | | RCP 2.6 | | | WTD (*10$^{-3}$ m a$^{-1}$) RCP 4.5 | | | RCP 2.6 | | | Summer ST(*10$^{-1}$ °C a$^{-1}$) RCP 4.5 | | | RCP 2.6 | | |
|---|---|---|---|---|---|---|---|---|---|---|---|---|---|---|---|---|---|---|---|---|---|---|---|---|
| 1 | **-41.8** | ± | 2.7 | **-44.2** | ± | 1.84 | **-555** | ± | 24 | **-487** | ± | 70 | **0.13**[1] | ± | 0.34 | 0.04 | ± | 0.05 | **0.20** | ± | 0.02 | **0.03**[1] | ± | 0.07 |
| 2 | **-39.8** | ± | 3.8 | **-40.3** | ± | 3.59 | **-614** | ± | 36 | **-492** | ± | 59 | **0.08**[1] | ± | 0.34 | 0.17 | ± | 0.10 | **0.20** | ± | 0.03 | 0.02 | ± | 0.06 |
| 4 | **-42.1** | ± | 3.8 | **-38.3** | ± | 3.56 | **-564** | ± | 24 | **-476** | ± | 52 | 0.14 | ± | 0.30 | **0.25**[1] | ± | 0.14 | **0.20** | ± | 0.03 | 0.01 | ± | 0.06 |
| 5 | **-39.2** | ± | 1.7 | **-44.6** | ± | 7.34 | **-508** | ± | 23 | **-459** | ± | 84 | **0.12**[1] | ± | 0.28 | 0.15 | ± | 0.01 | **0.21** | ± | 0.02 | 0.01 | ± | 0.06 |
| 6 | **-44.4** | ± | 3.5 | **-32.7** | ± | 2.87 | **-549** | ± | 22 | **-371** | ± | 45 | 0.09 | ± | 0.27 | 0.23 | ± | 0.05 | **0.21** | ± | 0.03 | 0.02 | ± | 0.04 |
| 7 | **-43.9** | ± | 5.5 | **-38.0** | ± | 3.50 | **-559** | ± | 3 | **-402** | ± | 75 | 0.17 | ± | 0.13 | 0.14 | ± | 0.05 | **0.21** | ± | 0.03 | 0.03 | ± | 0.04 |
| 8 | **-37.8** | ± | 7.6 | **-44.2** | ± | 2.65 | **-553** | ± | 4 | **-474** | ± | 71 | 0.19 | ± | 0.11 | 0.08 | ± | 0.01 | **0.20** | ± | 0.03 | **0.03**[1] | ± | 0.06 |
| 9 | **-39.2** | ± | 8.7 | **-23.3** | ± | 4.80 | **-558** | ± | 14 | **-322** | ± | 41 | 0.10 | ± | 0.22 | 0.05 | ± | 0.18 | **0.20** | ± | 0.04 | **0.04**[1] | ± | 0.05 |
| 10 | **-43.8** | ± | 3.4 | **-32.5** | ± | 3.05 | **-539** | ± | 4 | **-400** | ± | 56 | 0.09 | ± | 0.19 | 0.16 | ± | 0.05 | **0.19** | ± | 0.04 | 0.03 | ± | 0.04 |
| 11 | **-49.0** | ± | 1.3 | **-31.9** | ± | 6.73 | **-547** | ± | 9 | **-302** | ± | 56 | 0.11 | ± | 0.28 | -0.05 | ± | 0.13 | **0.20** | ± | 0.03 | **0.04**[1] | ± | 0.05 |
| 12 | **-49.2** | ± | 2.7 | **-29.8** | ± | 5.57 | **-450** | ± | 29 | **-321** | ± | 51 | 0.06 | ± | 0.35 | 0.07 | ± | 0.15 | **0.20** | ± | 0.03 | **0.04**[1] | ± | 0.05 |
| 13 | **-46.5** | ± | 2.1 | **-41.4** | ± | 4.50 | **-474** | ± | 23 | **-387** | ± | 74 | 0.10 | ± | 0.23 | -0.05 | ± | 0.11 | **0.22** | ± | 0.04 | 0.04 | ± | 0.02 |
| 14 | **-42.6** | ± | 3.3 | **-48.2** | ± | 6.26 | **-533** | ± | 9 | **-434** | ± | 62 | **0.02**[1] | ± | 0.34 | -0.01 | ± | 0.01 | **0.20** | ± | 0.03 | 0.04 | ± | 0.03 |
| 15 | **-31.7** | ± | 3.8 | **-44.1** | ± | 7.56 | **-587** | ± | 21 | **-420** | ± | 42 | **-0.05**[1] | ± | 0.29 | 0.01 | ± | 0.9 | **0.20** | ± | 0.03 | 0.03 | ± | 0.00 |
| 16 | **-43.3** | ± | 2.5 | **-40.0** | ± | 8.78 | **-531** | ± | 9 | **-391** | ± | 69 | **-0.13**[1] | ± | 0.33 | 0.04 | ± | 0.20 | **0.20** | ± | 0.03 | 0.02 | ± | 0.05 |
| 17 | **-41.0** | ± | 4.3 | **-26.7** | ± | 7.10 | **-466** | ± | 19 | **-242** | ± | 41 | **-0.03**[1] | ± | 0.37 | 0.02 | ± | 0.16 | **0.20** | ± | 0.03 | 0.03 | ± | 0.04 |
| 18 | **-41.1** | ± | 4.3 | **-24.3** | ± | 5.68 | **-400** | ± | 24 | **-234** | ± | 40 | 0.07 | ± | 0.22 | 0.01 | ± | 0.08 | **0.20** | ± | 0.03 | 0.03 | ± | 0.04 |
| 19 | **-28.8** | ± | 4.5 | **-10.6** | ± | 3.49 | **-342** | ± | 11 | **-112** | ± | 13 | 0.02 | ± | 0.27 | **-0.14**[1] | ± | 0.15 | **0.23** | ± | 0.04 | 0.04 | ± | 0.02 |



**Table A5.** Linear trends and their uncertainty related to climate models of $CH_4$ fluxes, soil respiration, summer WTL and summer soil temperature in the Finnish counties in continuous cover forestry peatlands. The values show Sen's slope parameter averaged over the climate models and their variability. Bolded values indicate significant trends (p <0.05) as estimated by the Mann-Kendall trend test. [1] indicates only one model showed a significant trend, [2] indicates that two models showed a significant trend.

| location | $CH_4$ (*$10^{-2}$) kg ha$^{-1}$ a$^{-1}$ RCP 4.5 | RCP 2.6 | Fast decaying pools kg ha$^{-1}$ a$^{-1}$ RCP 4.5 | RCP 2.6 | WTD (*$10^{-3}$ m a$^{-1}$) RCP 4.5 | RCP 2.6 | Summer ST (*$10^{-1}$ °C a$^{-1}$) RCP 4.5 | RCP 2.6 |
|---|---|---|---|---|---|---|---|---|
| 1 | 0.09 ± 0.32 | 0.01 ± 0.28 | **74** ± 10 | **94** ± 7 | **0.57**[1] ± 0.56 | 0.18 ± 0.24 | **0.21** ± 0.02 | **0.03**[1] ± 0.07 |
| 2 | 0.05 ± 0.32 | 0.17 ± 0.16 | **28**[2] ± 3 | **100** ± 1 | 0.33 ± 0.63 | 0.41 ± 0.21 | **0.21** ± 0.03 | **0.02**[1] ± 0.07 |
| 4 | 0.10 ± 0.26 | 0.17 ± 0.08 | **85** ± 11 | **104** ± 11 | 0.51 ± 0.44 | 0.60 ± 0.18 | **0.20** ± 0.02 | 0.01 ± 0.06 |
| 5 | 0.09 ± 0.29 | 0.16 ± 0.27 | **110** ± 20 | **105** ± 5 | **0.57**[1] ± 0.44 | 0.46 ± 0.16 | **0.21** ± 0.02 | 0.02 ± 0.07 |
| 6 | 0.05 ± 0.24 | 0.18 ± 0.25 | **85** ± 14 | **114** ± 18 | 0.53 ± 0.32 | 0.70 ± 0.47 | **0.21** ± 0.02 | 0.03 ± 0.04 |
| 7 | 0.09 ± 0.15 | 0.04 ± 0.33 | **77** ± 13 | **110** ± 9 | 0.56 ± 0.33 | 0.43 ± 0.18 | **0.21** ± 0.03 | 0.03 ± 0.05 |
| 8 | 0.16 ± 0.16 | 0.05 ± 0.27 | **83** ± 14 | **102** ± 5 | 0.68 ± 0.26 | 0.20 ± 0.10 | **0.21** ± 0.03 | **0.03**[1] ± 0.06 |
| 9 | 0.20 ± 0.17 | 0.00 ± 0.44 | **87** ± 9 | **120** ± 1 | 0.65 ± 0.30 | **0.72**[1] ± 1.01 | **0.20** ± 0.04 | **0.05**[1] ± 0.06 |
| 10 | 0.18 ± 0.23 | 0.06 ± 0.18 | **107** ± 14 | **127** ± 3 | 0.61 ± 0.32 | 0.42 ± 0.02 | **0.19** ± 0.04 | 0.02 ± 0.05 |
| 11 | 0.15 ± 0.29 | -0.12 ± 0.11 | **104** ± 10 | **159** ± 10 | **0.61**[1] ± 0.49 | -0.14 ± 0.45 | **0.20** ± 0.04 | **0.04**[1] ± 0.05 |
| 12 | **0.31**[1] ± 0.35 | -0.06 ± 0.15 | **175** ± 27 | **161** ± 10 | **0.63**[1] ± 0.60 | 0.04 ± 0.37 | **0.20** ± 0.03 | 0.04 ± 0.05 |
| 13 | **0.18**[1] ± 0.26 | -0.12 ± 0.12 | **152** ± 23 | **134** ± 3 | **0.44**[1] ± 0.39 | -0.10 ± 0.17 | **0.22** ± 0.04 | 0.04 ± 0.04 |
| 14 | **0.06**[1] ± 0.38 | -0.09 ± 0.03 | **135** ± 17 | **122** ± 0 | **0.32**[1] ± 0.69 | 0.02 ± 0.06 | **0.20** ± 0.02 | 0.03 ± 0.04 |
| 15 | -0.07 ± 0.31 | -0.07 ± 0.11 | **106** ± 10 | **128** ± 6 | 0.10 ± 0.48 | 0.01 ± 0.21 | **0.19** ± 0.02 | 0.01 ± 0.04 |
| 16 | **0.02**[1] ± 0.40 | -0.05 ± 0.19 | **139** ± 16 | **132** ± 4 | **0.18**[1] ± 0.80 | -0.01 ± 0.36 | **0.19** ± 0.02 | 0.02 ± 0.05 |
| 17 | **0.33**[1] ± 0.34 | -0.05 ± 0.07 | **177** ± 26 | **181** ± 6 | **0.80**[1] ± 0.81 | -0.12 ± 0.22 | **0.19** ± 0.03 | 0.03 ± 0.04 |
| 18 | **0.34**[2] ± 0.40 | -0.06 ± 0.03 | **196** ± 33 | **190** ± 7 | **0.79**[1] ± 0.60 | 0.03 ± 0.15 | **0.19** ± 0.03 | 0.03 ± 0.04 |
| 19 | **0.30**[1] ± 0.14 | -0.11 ± 0.08 | **215** ± 20 | **261** ± 5 | **0.89**[1] ± 0.40 | -0.31 ± 0.25 | **0.23** ± 0.04 | 0.04 ± 0.02 |





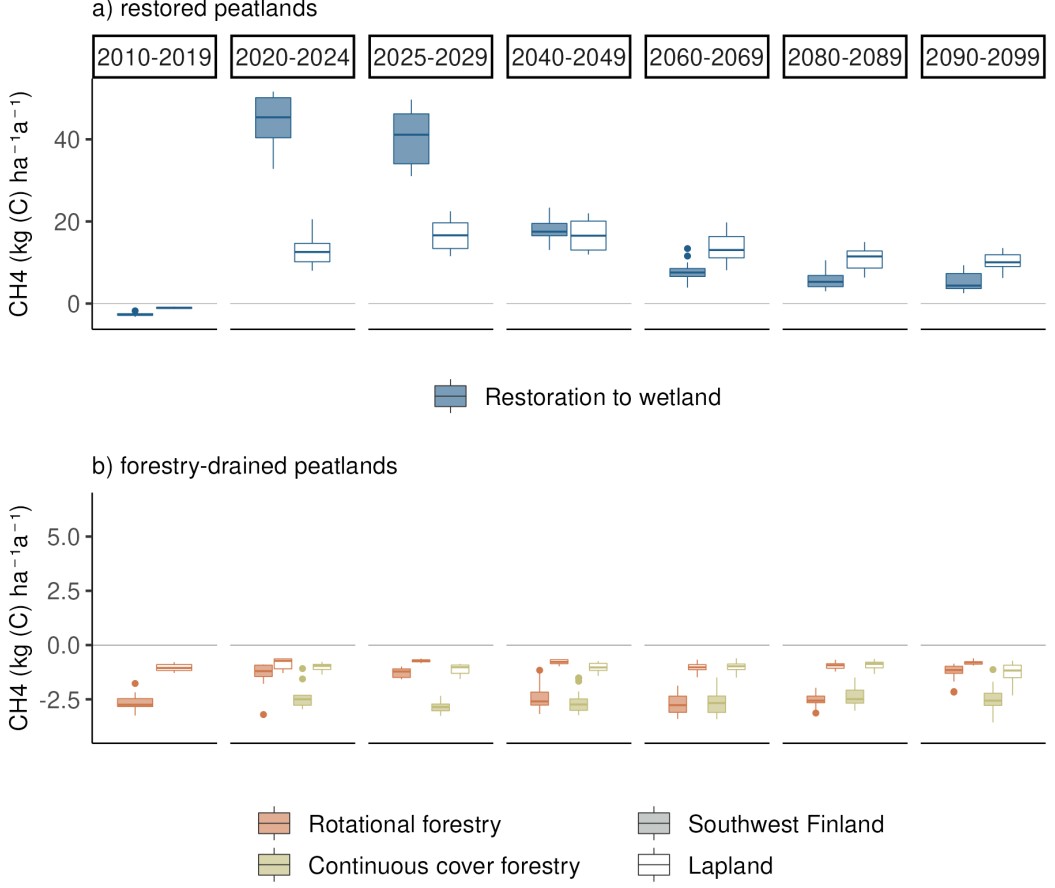

**Figure A1.** Average annual CH4 flux in (a) restored peatlands and (b) under forest management scenarios (rotational and continuous cover forestry) during the 21st century under RCP scenario 2.6 in Southwestern Finland and Lapland. Each boxplot shows variation in the CH$_4$ flux created by yearly variation and differences in the three climate models. The first column (2010-2019) shows the situation in a mature forest before the beginning of the different management scenarios. The next two columns (2020-2024 and 2025-2029) show the situation during the first decade of the different management scenarios and the last three columns (2040-2049, 2060-2069, 2080-2089 and 2090-2099) show the development of CH4 flux during the later decades.





**Figure A2.** The impact of forest management on summer WTL, soil temperature (-22 cm below ground) and fast-decaying carbon pools under RCP scenario 4.5. Each boxplot shows variation created by yearly variation and differences in the three climate models. The first column (2010-2019) shows the situation in a mature forest before the beginning of the different management scenarios. The next two columns (2020-2024 and 2025-2029) show the situation during the first decade of the different management scenarios and the last three columns (2040-2049, 2060-2069, 2080-2089 and 2090-2099) show the development of the variables during the later decades.



## Appendix B: JSBACH-FOM description

### B1 Model description

A version of JSBACH4 has recently been developed that can account for a distribution of forest age classes within the forest PFTs (Nabel et al., 2020). The model we used, JSBACH-FOM, also includes the age for forest PFTs, but it was implemented into a JSBACH3 version. In addition to the ability to account for the age of forest PFTs, the model differs from earlier JSBACH3 versions in the way the maximum LAI is treated. The maximum LAI that can be reached during the growing season has previously been PFT dependent, but constant. In the JSBACH-FOM the maximum LAI is dependent on the available leaf biomass, needed for simulating forest regrowth and age.

In our work the phenology was calculated using the Logistic Growth Phenology (LoGro-P) model (Böttcher et al., 2016). The LoGro-P model provides the seasonal development of LAI for each PFT based on the temperature and soil moisture. The LAI is further limited by a maximum LAI, which is either prescribed for each PFT, or as in our case dependent on the available leaf carbon for the forest PFTs. Other PFT dependent parameters are the phenology type (e.g. evergreen and grass) and specific leaf area (SLA).

The calculation of the growing forest starts from the total vegetation carbon. The number of trees per area, or stem number, is calculated from the total vegetation carbon assuming that the forests are in self-thinning state. The biomass per individual can then be used together with allometric relationships to derive the maximum LAI of the forest. The implementation of the forest growth is based on maximum LAI. The setup of the model parameters for our simulations is explained in detail in section B2. The forest ageing is done at the beginning of each year. The harvest is done for each tile when the rotation time is reached. Once the criteria is reached, the forest is clear-cut at the beginning of the year, and the age and size of the trees restarts from zero. The carbon pools are redistributed due to harvesting. The harvested carbon taking 77% of the above ground woody pool, is removed from the calculations. Furthermore, all green and reserve carbon, both above and below ground, go into the Yasso AWEN litter pools according to predefined fractions. The below ground woody carbon is distributed into the below ground litter pools. In addition, a fraction of the above ground woody carbon, i.e. the slash fraction, is distributed into the above ground litter pools. The slash fraction equals 23% of the above ground woody carbon pool and accounts for the damage and small woody residues left in the forest during harvest.

### B2 Model setup

The standard setup of land surface parameters for extratropical coniferous forest was used, with some modifications, to represent Scots pine. The specific leaf area (SLA) was set to 61.62 cm$^2$ g$^{-1}$ according to (Goude et al., 2019). Competition for resources between trees in a stand results in self-thinning. The pine forest in the model is assumed to be in a self-thinning state. The parameters needed for describing the forest growth were derived from published expressions for self-thinning and allometric relationships for Scots pine. Self-thinning of even-aged stands can be modelled using a relationship between the




quadratic mean diameter and the number of trees per area (equation B1) according to (Reineke, 1933).

$$ln(N) = p - q \cdot ln(D_g) \tag{B1}$$

where $N$ is the stem number (ha$^{-1}$) and $D_g$ is the mean diameter at breast height (dbh) in cm weighted with the basal area, i.e. the quadratic mean diameter. Each tile represents only one age-class of trees, and the model has only one diameter per age-class, but the quadratic mean equals the arithmetic mean when the variance is zero. The intercept ($p$) and slope ($q$) of the log-log relationship were obtained from (Hynynen, 1993), 12.669 and -1.844, respectively. (Hynynen, 1993) used data for Scots pine from 19 unthinned, even-aged and monospecific plots in Finland to derive the parameters $p$ and $q$. Only plots where no extensive natural disturbance had occurred during the study period were included. The data was collected between 1924 and 1989, on average six measurements over 38 years. In the JSBACH-FOM the self-thinning expression (equation B2) relates the stem number and the biomass per unit area.

$$ln(BM_{veg}) = \alpha_{nr\_ind} + \beta_{nr\_ind} \cdot ln(N) \tag{B2}$$

where $BM_{veg}$ is the vegetation carbon at maximum green pool (kg m$^{-2}$). In order to derive the slope ($\beta_{nr\_ind}$) and intercept ($\alpha_{nr\_ind}$) allometric relationships between the dbh and the biomass of various above ($AB$) and below ground ($BG$) components are needed.

$$BM_{ind} = AG + BG \tag{B3}$$

The biomass per area is obtained from the biomass of a single tree, $BM_{ind}$ (kgm$^{-2}$), and the stem number.

$$BM_{veg} = BM_{ind} \cdot N/10000 \tag{B4}$$

We considered allometric relationships for pine based on data from Finnish sites compiled by (Zianis et al., 2005). A summary is given in table B1. First the total biomass is obtained as a function of dbh by summing the AG and BG biomass according to equation B3. When both the stem number and the biomass is expressed as a function of dbh the coefficients $\alpha_{nr\_ind}$ and $\beta_{nr\_ind}$ in equation B2 can be obtained by plotting $ln(BM_{veg})$ against $ln(N)$. We used four allometric relationships for the AB, and and two for the BG biomass, which gave eight different relationships between biomass per area ($BM_{veg}$) and stem number ($N$). These are plotted in figure B1, grey lines. In addition we used allometric relationships by (Marklund, 1988) based on data from Sweden, to derive one more relationship, cyan line in figure B1.

The relationships are plotted only for the valid range of the original allometric relationships (given in table B1). There is an upper limit for the stem number in the plot, i.e. $ln(N) = 9.5$, based on a cut-off value in JSBACH-FOM, which is used to prevent an excess number of very small trees. From figure B1 it can be seen that the biomass based on (Marklund, 1988) fits within the range of the ones derived from data from Finnish plots. We selected the parameter values based on (Marklund, 1988) for the simulations due to the fact that the equations are based a large sample size, hundreds of felled trees from a large geographical area. In addition, the valid range of dbh (0-45 cm) is larger than for the other allometric relationships. (Marklund, 1988) also the provides relationships for both above and below ground biomass. In the final simulations we used a modified relationship, dashed blue line in figure B1.





**Table B1.** References for biomass equations that were considered in this publication (Biomass types: AG=above ground; ST=stem; CR=crown; BG=below ground; FL=foliage). Above ground biomass is composed of stem and crown biomass.

| Biomass | Equation | Remark | Reference |
|---------|----------|--------|-----------|
| AG | $AG = 18.779 - 4.328 \cdot D + 0.506 \cdot D^2$ | | (Briggs and Cunia, 1982) |
| AG | $AG = 7.041 - 1.279 \cdot D + 0.201 \cdot D^2$ | | (Briggs and Cunia, 1982) |
| AG | $ln(AG) = -3.2807 + 2.6931 \cdot ln(D)$ | Dominant trees over an age gradient | (Mäkelä and Vanninen, 1998) |
| AG | $ln(AG) - 2.3042 + 2.2608 \cdot ln(D)$ | Trees of different sizes in one age group | (Mäkelä and Vanninen, 1998) |
| AG | $AG = ST + CR$ | Sum of stem and crown biomass | (Marklund, 1988) |
| | $ST = -2.3388 + 11.3264 \cdot [D/(D+13)]$ | $0 < D/cm < 45$ | |
| | $CR = -2.8604 + 9.1015 \cdot [D/(D+10)]$ | $0 < D/cm < 45$ | |
| BG | $log(BG) = -1.967 + 2.458 \cdot log(D)$ | $7 < D/cm < 21.6$ ; $D_{root} > 1cm$ | (Mälkönen, 1974) |
| BG | $log(BG) = -1.89 + 2.74 \cdot log(D)$ | $4 < D/cm < 24$ | (Drexhage and Gruber, 1999) |
| BG | $ln(BG) = -3.3913 + 11.1106 \cdot [D/(D+12)]$ | $0 < D/cm < 45$ | (Marklund, 1988) |
| FL | $FL = 0.023 \cdot D + 0.015 \cdot D^2$ | Needles, twigs, and branches $D < 1cm$ | (Briggs and Cunia, 1982) |
| FL | $FL = -0.105 + 0.365 \cdot D + 0.01 \cdot D^2$ | Needles, twigs, and branches $D < 1cm$ | (Briggs and Cunia, 1982) |
| FL | $ln(FL) = -7.47 + 1.6975 \cdot ln(D)$ | | (Hakkila, 1991) |
| FL | $ln(FL) = -0.7714 + 0.9513 \cdot ln(D)$ | Dominant trees over an age gradient | (Mäkelä and Vanninen, 1998) |
| FL | $ln(FL) = -5.613 + 2.5804 \cdot ln(D)$ | Trees of different sizes in one age group | (Mäkelä and Vanninen, 1998) |
| FL | $ln(FL) = -3.7983 + 7.7681 \cdot [D/(D+7)]$ | $0 < D/cm < 45$ | (Marklund, 1988) |

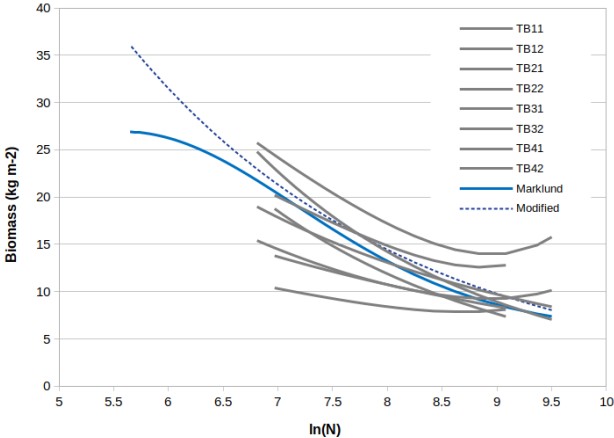

**Figure B1.** Biomass per area as a function of stem number. The lines have been cut-off based on the valid range of dbh, in addition the model has a maximum stem number defined as $ln(N)_{max} = 9.5$. (Marklund, 1988) is shown with a cyan line and modified Marklund with a dashed blue line with parameters $\alpha_{nr\_ind} = 15$ and $\beta_{nr\_ind} = -0.39$.





The JSBACH-FOM also requires the relationship of leaf carbon ($C_{leaf}$) and the biomass of a tree ($BM_{ind}$).

$$ln(C_{leaf}) = \alpha_{leaf} + \beta_{leaf} \cdot ln(BM_{ind}) \tag{B5}$$

Again we used the allometric relationships for AG and BG ground biomass in table B1 to derive the total biomass of a tree,
which gave eight descriptions for the tree biomass ($BM_{ind}$). In addition we had five expressions describing the foliage biomass
as a function of dbh, based on data from Finnish sites (table B1). The foliage biomass was divided by two to obtain $C_{leaf}$. These
expressions were used to derive 40 different relationships between leaf carbon ($C_{leaf}$) and tree biomass ($BM_{ind}$) according to
equation B5. These are plotted in figure B2 with grey lines. The relationships are again plotted only for the valid range of the
original allometric relationships (given in table B1). We also used allometric relationships by (Marklund, 1988) based on data
from Sweden, to derive one set of $\alpha_{leaf}$ and $\beta_{leaf}$, cyan line in figure B2. We decided to use these in the JSBACH-FOM due to
the wide range of dbh where the relationships are valid. According to figure B2 the relationship from the (Marklund, 1988) data
fits within the range of the ones derived from data from Finnish plots. Based on test simulations we modified the (Marklund,
1988) coefficients for the final simulations, dashed blue line in figure B2. In the model the maximum LAI is calculated from
leaf carbon per tree using the SLA. The increase of the maximum LAI is stopped when then the increase in maximum LAI per
year is less than 1 %.

The parameters derived from the (Marklund, 1988) biomass equations for pine, described in section B2, were adjusted
manually in order to get a better agreement between simulated and observed GPP. The relationships are plotted in figure B1
and B2.

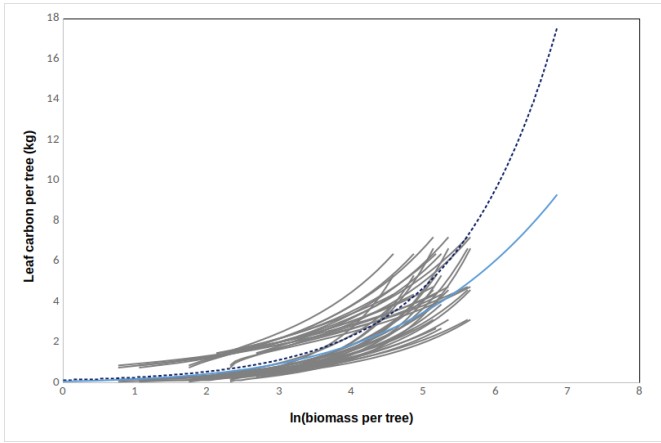

**Figure B2.** Relationships describing the leaf carbon as a function of total biomass. (Marklund, 1988) is shown with a cyan line and modified
Marklund with a dashed blue line with parameters $\alpha_{leaf}$ = -2.0 and $\beta_{leaf}$ = 0.71.



**Table B2.** Summary of comparison of model results with observations for Hyytiälä and Sodankylä. Hyytiälä $CO_2$ fluxes are average of 2002–2007. All-sided LAI divided by 2.57 to obtain one-sided LAI

| | Hyytiälä (40 years old forest) | | |
| Parameter | model | observation | reference |
| --- | --- | --- | --- |
| Stem number (ha$^{-1}$) | 5000 | 1400 | (Kolari et al., 2009) |
| LAI (m$^2$ m$^{-2}$) | 2.5-3.2 | [a)]2.3 | (Palmroth and Hari, 2001) |
| Tree biomass (kg m$^{-2}$) | 12 | 7.2(10.4) | (Ilvesniemi et al., 2009) |
| Foliage biomass (kg m$^{-2}$) | 0.64 | 0.45 | (Kolari et al., 2009) |
| Litter flux (g$_C$ m$^{-2}$ a$^{-1}$) | 390 | 232–294 | (Ilvesniemi et al., 2009) |
| GPP (g$_C$ m$^{-2}$ a$^{-1}$) | 950 | 1051 | (Kolari et al., 2009) |
| NEE[b)] (g$_C$ m$^{-2}$ a$^{-1}$) | -200 | -209 | (Kolari et al., 2009) |
| R$_e$ (g$_C$ m$^{-2}$ a$^{-1}$) | -750 | -837 | (Kolari et al., 2009) |





*Author contributions.*   VT, TM and TA designed the study; TM, LB, MR, AL and XL set up the model; VT and TN performed the simulations; PO, KM, RH, MP, JA and RL provided the field-measured data; KM, AL, RM and TA secured the funding for the work; VT prepared the manuscript with contributions from all co-authors.

*Competing interests.*   The authors declare that they have no conflict of interest.

*Acknowledgements.*   We thank Anna-Liisa Granqvist, Jyrki Jauhiainen, Liisa Jokelainen, Päivi Mäkiranta, Meeri Pearson and Timo Penttilä
for assisting in the collection of field data. We thank Kim Naudts and Julia Nabel for their work on JSBACH-FOM. This research was funded by the EU-H2020 VERIFY (776810), FIRI - ICOS Finland (345531), ICOS-ERIC (281250), Academy of Finland Center of Excellence (272041), Academy of Finland Grant no. 337552 (Flagship), 325169 (METNET) and 350184/341753 (RESPEAT), Strategic Research Council of Finland (SOMPA project, decision no 336570, 336573 and 312932), Ministry of Agriculture and Forestry Finland Grant no. 4400T-2105 (TURNEE), EU-Horizon Eye-Clima (101081395), EU-Alfawetlands (101056844), EU-Wethorizons (101056848), EU LIFE21-
CCM-LV-LIFE PeatCarbon – 101074396 and LIFE18 CCM/LV/001158 LIFE OrgBalt. In the Natural Resources Institute Finland (Luke), the study has been done with affiliation to the UNITE Flagship funded by the Finnish Research Council (decision no 337655).



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
