# Peer review of "Future methane fluxes of peatlands are controlled by management practices and fluctuations in hydrological conditions due to climatic variability"

_EGUsphere, 2023_

## Author Comment (AC1)

We thank the anonymous referee #1 for their valuable and constructive feedback. Their detailed and professional review was very useful and important for improving the manuscript. Here below, we provide a point-by-point response letter addressing the comments. Our responses are in blue and the line numbers (L) refer to the manuscript. The cited references are provided at the end of the letter. We thank you for your time and effort. Stay safe and take care.

On behalf of all the authors,
Sincerely,
Vilna Tyystjärvi

This manuscript aims to model methane fluxes over peatlands under different management and climate scenarios. In general, the manuscript is well written and easy to read. It is important to understand variation in methane fluxes considering its potent nature.

However, I would like the authors to discuss one important aspect of such modelling studies. From the manuscript, I can see that this is a quite complex modelling exercise involving lot of coupled models and formulations (e.g. for biomass). A lot of published papers are cited to support the modeling framework that was adopted. In reality, one cannot read all those papers, even to understand the models on a stand-alone basis. Understanding the complex coupling between them is even more challenging. There could be significant uncertainties in model outputs even for a stand alone model. Given this, how confident one can be in the results obtained (given the limited number of observation sites) and further, how others can replicate such studies? Even a minor change in model parameters can lead to some major changes in the results!

This is an excellent point and one that we agree should be discussed in more detail in the paper. It is true that these types of models are very complicated because they aim to describe a vast number of different processes to, more or less, accurately describe the functioning of the whole land surface and atmosphere interactions. Thus, describing even the basic functioning of the main model, JSBACH, here in a way that would help the reader understand all its relevant aspects is not viable. Furthermore, sufficient calibration data to evaluate these models is limited which then also increases the level of uncertainty in the models. On the other hand, simulating possible changes in complex ecosystem processes at large scales and into the future requires these types of models, so they are a necessary tool to help us understand the possible impacts of human actions on ecosystems.

We will add a point about these issues and compromises related to complex modelling systems to section 1. In section 4.3, we will discuss further the uncertainty related to complex modelling systems. Considering the uncertainty, while we haven't calibrated each parameter in this work separately (which indeed would not be possible in this case), the most important aspects of the work have been tested in previous papers which does reduce the risk of increasing uncertainty with poor parameter choices. The forestry-related parameters in JSBACH-FOM have been calibrated against forest measurements in Nordic countries as shown in Appendix B. Methane-related parameters have been tested in Raivonen et al (2017) and further tested in Li et al to better suit these types of peatland forests. Furthermore, research done by Mäkelä et al. (2019) indicates that the largest source of uncertainty in climate scenario driven simulations of carbon balances in LSMs comes from climate-related uncertainty rather than from parameter

uncertainty. This we have taken into account by considering multiple climate models and emission pathways. Nonetheless, it is true that replicating this study with for example a different modelling set up could lead to different results as each model describes relevant processes and parameters somewhat differently. This is also an important aspect to consider and highlights the need to use various models and setups to understand model-related uncertainty.

In addition to this, I believe that the peatlands scenarios modelled in this manuscript are hypothetical in nature (please correct me if I am wrong). Then how beneficial this study will be?

These peatlands are hypothetical in the sense that we haven't tried to replicate any existing peatland site but rather tried to estimate the average responses to climate and land management at larger scales. However, the peatland characteristics describe average conditions in northern peatlands and the forestry scenarios are based on management types used in Finland (Juutinen et al. 2021), albeit in a simplified way to allow them to be simulated in the model. In this sense, they are not purely hypothetical and thus provide a way to understand the possible impacts of combined climate change and land use changes on northern managed peatlands. Furthermore, the aim of this paper is not to be a tool for a single forest owner but rather to help us estimate regional changes in the future. Simulating this at site level would make it more challenging to distinguish regional differences from differences created by site-specific characteristics.

In a more generic manner, given the increase in computational resources, anyone can assume some scenarios, model it and try to publish a paper. How such modelling studies will benefit the scientific community as a whole? If such studies are not justified strongly, this can potentially lead to hundreds of modelling papers. (To the authors: This is a more generic comment I have in my mind seeing the proliferation of modelling studies in the scientific literature. Some of the modelling studies do not even consider the uncertainties or performance of the underlying datasets but make strong claims. Please do not take this as a personal comment. You may please provide justifications to show that your manuscript is not just any one modelling study but it makes a real strong case).

We agree that modelling studies should carefully take into account the reasoning and implementation of model simulations and scenarios as well as the reliability and usability of the results. Considering the importance of drained peatlands and their management options in Europe, and particularly in Finland, we consider that it's extremely important to understand the impacts of these options now and in the future. The scenarios we have used are the main management options currently used in Finnish peatland forestry, with one "business as usual" option (clearcutting) and two climatologically better options (restoration and partial harvesting). The model system used is complex enough to simulate all the relevant processes, as well as describe changes in climate and land management. Thus, we consider the reasoning of this modelling study to be well justified. While this doesn't likely provide a complete final answer on the functioning of drained peatlands, it does provide useful information on how these regions may change and what should be considered in future studies of these areas. We will emphasise this point in section 1.

Page 3, line 75: What is a nutrient-rich and nutrient poor peatland? How do they differ in terms of functioning?

Nutrient-rich, or minerotrofic, peatlands receive their water, as well as nutrients, from the surrounding land either from groundwater or surface water whereas nutrient-poor, ombrotrophic, peatlands receive water solely from precipitation. This in turn influences vegetation and carbon dynamics in the peatlands with peatland forests on nutrient-poor peatlands possibly staying as carbon sinks (Lohila et al 2011). We will expand on these terms in section 1 to clarify the differences.

Page 4, section 2.1: The JSBACH model uses PFT's to represent the vegetation functions in the model. Since this is a case study specific to Finland, are the vegetation PFTs prescribed in the model correspond to the vegetation found in Finland. For example, a same type of vegetation may exhibit different growing phases/phenology etc. depending on the site. Hence, to learn about how CH4 fluxes vary in Finland, the vegetation types and the peatland characteristics coded in the model should match the ground conditions. How to ensure this?

The PFTs used in this study describe well the average conditions in Finnish peatland forests and wetlands of which a considerable portion is coniferous forests. Therefore, the phenology in the PFT is appropriate for Finnish peatland forests. Furthermore, our implementation takes into account regionally relevant aspects of vegetation properties such as the delayed effect of temperature for photosynthetic activity in spring (Mäkelä et al 2019). The forest growth has been matched to measurements done in boreal forests. The parameters describing soil processes and methane production have been calibrated to match Finnish peatlands. This means that the results may not describe any specific ground conditions in Finland precisely, as fine-scale variation in both vegetation and soil properties is often considerable and unfruitful to consider in this type of study, but they do describe well the average conditions throughout Finland.

Section 2.1.1, line 100 (and several other places): Reference to a paper under preparation is given. Please do not cite a paper under preparation to support your work. How one can check the scientific validity of the work that is under preparation? Technically, all the novelties in the manuscript under preparation must be presented here.

This is completely true. The paper that we cited was submitted in autumn 2023 while we were preparing the manuscript and this has now been corrected in our manuscript. The paper cited is available as a preprint here: https://papers.ssrn.com/sol3/papers.cfm?abstract_id=4170450.

Page 7: Line 176: Can you give a mean value of the carbon stock as simulated by the model after the spinup, for different regions in Finland, along with standard deviation?

This is a good suggestion, we will add a table to section 3.1.

Section 2.2: The management practices assumed for the modelling simulations adhere to the actual ground conditions followed in Finland?

While the simulated management practices are somewhat simplified from real-life practices, they do describe currently used management practices in Finland.

Section 2.3: Can you provide a map of Finland with the different regions marked (as the region names are used frequently in the paper, it is difficult to attach a location to them) and showing the location of the flux measurement sites?

We will add the suggested map to section 2.2.

Section 3.1: The first line is too generic and qualitative in describing the model's performance. Please provide some useful numerical metrics to understand the same. From Figure 2, I can see significant differences between the model simulated results and the ground observed fluxes.

The aim of the first line was to be generic and qualitative as it starts a paragraph. However, we will modify it to be more exact. We will also add numerical metrics, median and standard deviation, to either the figure or a separate table to provide more information on the model performance. However, we would like to point out that the aim of this comparison is not to show that the model corresponds precisely to the measurements. That would be impossible as the model has not been trained to describe these exact peatland sites. The useful information in figure 2 is that 1) the simulated and measured fluxes are in the same scale, i.e. the model doesn't drastically over- or underestimate the fluxes, and 2) the dynamics in comparison to changes in soil temperature (2a) and water-table depth (2b) are similar in measurements and the model, e.g. when WTD rises, methane sinks weaken. We will explain this in more detail in section 3.1 and explain in section 4.3 further how the measurements and model results can be compared.

Page 17: line 329: Is the amount of methane contained in the fast decaying pools fixed? Then only an initial overestimation can lead to later underestimation. Won't there be any CH4 cycle to replenish this pool? Also, in the same line, the word 'latter' should be changed to 'later'.

The amount of methane in the fast-decaying pools is not fixed. As explained in 2.1.4., HIMMELI simulates also the production of methane, in addition to oxidation, transport etc. The word "latter" is used to describe for example something related to the end of something or to the second part of two groups, in this case, the second half of the century (https://www.merriam-webster.com/dictionary/latter).

Page 4, line 91: In this work....simulations with only ONE PFT per site (the word one is missing).

Thank you for noticing that!

References:
Juutinen, A., Shanin, V., Ahtikoski, A., Rämö, J., Mäkipää, R., Laiho, R., ... & Saarinen, M. (2021). Profitability of continuous-cover forestry in Norway spruce dominated peatland forest and the role of water table. *Canadian Journal of Forest Research*, 51(6), 859-870. https://doi.org/10.1139/cjfr-2020-0305

Li, X., Markkanen, T., Korkiakoski, M., Lohila, A., Leppänen, A., Aalto, T., ... & Raivonen, M. Modelling the alternative harvesting effects on soil CO2 and CH4 fluxes from peatland forest by JSBACH-HIMMELI model. Available at SSRN 4170450. https://dx.doi.org/10.2139/ssrn.4170450

Lohila, A., Minkkinen, K., Aurela, M., Tuovinen, J. P., Penttilä, T., Ojanen, P., & Laurila, T. (2011). Greenhouse gas flux measurements in a forestry-drained peatland indicate a large carbon sink. *Biogeosciences*, 8(11), 3203-3218. https://doi.org/10.5194/bg-8-3203-2011

Mäkelä, J., Knauer, J., Aurela, M., Black, A., Heimann, M., Kobayashi, H., ... & Aalto, T. (2019). Parameter calibration and stomatal conductance formulation comparison for boreal forests with adaptive population importance sampler in the land surface model JSBACH. *Geoscientific Model Development*, 12(9), 4075-4098. https://doi.org/10.5194/gmd-12-4075-2019

Raivonen, M., Smolander, S., Backman, L., Susiluoto, J., Aalto, T., Markkanen, T., ... & Vesala, T. (2017). HIMMELI v1. 0: HelsinkI Model of MEthane buiLd-up and emIssion for peatlands. *Geoscientific Model Development*, 10(12), 4665-4691. https://doi.org/10.5194/gmd-10-4665-2017

---

## Author Comment (AC2)

We thank the anonymous referee #2 for their valuable and constructive feedback. Their detailed review was very useful and important for improving the manuscript. Here below, we provide a point-by-point response letter addressing the comments. Our responses are in blue and the line numbers (L) refer to the manuscript. The cited references are provided at the end of the letter. We thank you for your time and effort. Stay safe and take care.

On behalf of all the authors,
Sincerely,
Vilna Tyystjärvi

This modeling study investigates how the management of peatlands may change methane fluxes and how this could change in the future. The authors have made an enormous modeling effort to combine different modules in order to simulate water level fluctuations and methane emissions and sinks on peatlands. This is precisely why it is essential to describe the model in more detail, because almost all the equations for all the assumptions are missing. I was not able to recapitulate how things are related and which are the most important equations. The figure helps, but equations are definitely needed too. I even thought that modeling effort could be a separate technical article.

We fully understand that the model description is difficult to understand. However, these types of global land surface models are typically, as well as in the case of JSBACH, the result of years or decades of work by often a large group of scientists. This means that the resulting models are then extremely complex and understanding their functioning often requires considerable effort even from experienced modellers (e.g. Fisher et al 2020). Thus, a more detailed description is not viable for this manuscript and would likely not largely help in understanding the modelling effort. Furthermore, all model components, with their relevant equations, have been described in detail in previous, cited literature. We have described in detail the changes done for this manuscript (see appendix B). The combination of JSBACH and HIMMELI has been described in Li et al.

As this is a very local and even management-specific application of this model combination, a more comprehensive evaluation is required. The authors only show a very superficial evaluation on flux towers, not even showing measurements and simulations on a specific site.

We agree that this is a management-specific application of the model but we would like to emphasise that this is not a local modelling study but a regional one. The difference is that in a local application, we would indeed show measurements and simulations on a specific site and would try to describe accurately the conditions on a specific site, whereas in a regional simulation study, we aim to describe the average conditions in a specific region and not the specific conditions on a site. Due to this, comparing exact site measurements and model results would not be appropriate as they do not describe the same things. What the comparison of model results to chamber measurements, *not* flux towers, shows, is that 1) the simulated and measured fluxes are in the same scale, i.e. the model doesn't drastically over- or underestimate the fluxes, and 2) the dynamics in comparison to changes in soil temperature (2a) and watertable depth (2b) are similar in measurements and the model, e.g. when WTD rises, methane sinks weaken. More comprehensive evaluation of the model processes to flux towers can be found for example in Mäkelä et al (2019) or Raivonen et al. (2017).

The only conclusion I can take away is that the model is within the range of the measurements, which is relatively easy to achieve.

As mentioned above, another, and a critical take-away in our opinion is also that the model results and measurements show similar responses to temperature and water table depth.

The study does not show that specific points can be reproduced, nor does it show that management option can reproduce the behavior that was observed. I would also like to see if the changes in water table height are in the right range. I know this is not easy to observe, but it seems easier to do on a local scale.

As this is not a local study design, the aim was not to show that specific points can be reproduced. This would require tuning the parameters of the model to fit specific peatland sites which would then make it more difficult to compare the results regionally, which was the intention in this paper. We do not quite agree with the referee that the management options cannot reproduce the observed behaviour. It is true that there is more variation in the measurements, which is to be expected as there is more fine-scale variation in the peatland sites but on average, they show similar behaviour. The manuscript by Li et al, however, shows more precise comparisons of model results to measurements on the local scale. Concerning the water table depth, figure 2b shows the range in forested peatlands but we will add information concerning WTD on restored peatlands to the figure. This would, indeed, be easier on a local scale, and it would be interesting in the future to study this further with a local study design.

A general remark, I find it really hard to believe that peatlands are not methane sources from what I know. Anoxic decomposition definitely leads to higher methane concentration in soils. What process drives the sink behavior? Diffusion would merely equalize the concentration in the atmosphere and in the soil. Yes, atmospheric methane concentration will increase, but I couldn't find out i if this is accounted for. The present study pays particular attention to this.

It is true that anoxic decomposition leads to methane production in peat. However, when a peatland is drained, the water level drops and the surface peat layer gets oxic. In it, the methanogenic activity then decreases and methanotrophic activity increases. Methanotrophic microbes living in soils oxidise methane to $CO_2$ (e.g. Hornibrook et al. 2009). In the drained oxic surface peat, both methane produced in the anoxic peat below the water level and atmospheric methane is oxidised. This can turn the peatland into a small sink of atmospheric methane. Observations have been reported, for example, in Glenn et al. (1993), Korkiakoski et al. (2020), Ojanen et al. (2010) and Roulet et al. (1993). We will add a more detailed explanation of this in the introduction. Concerning the atmospheric concentration of methane, we have not considered this in this study but this is something that should possibly be considered in future research. At this moment, we do not have sufficient data of the changes in atmospheric methane, that is could be reliably used.

*Methods*: Which PFT's are meant to be peatland-PFT's and which plants do they represent?

We agree that the sentence in lines 91-92 is somewhat confusing and will rewrite it. Peatland-PFT should be wetland-PFT and it represents vegetation that typically grows on natural wetlands.

*Experiment design*: If you would run the model for longer than 10000 years would the soil carbon pool be different? Is the model able to reach an equilibrium?

The model might reach equilibrium if it was run for considerably longer than 10 000 years. Similarly, actual peatlands might reach equilibrium at some point, if left undisturbed but as all peatlands in Finland have started after the last glacial period ~10 000 years ago, we haven't yet reached that point and therefore, have also not aimed to simulate an equilibrium state in carbon accumulation.

And did you get an discontinuity in 1951 when switching to JSBACH-FOM? I'm not sure if it is even possible to change from one model to another.

There was no discontinuity in 1950 when switching to JSBACH-FOM and we are not entirely sure why this switch shouldn't be possible. The two model versions are not completely different models, just different versions of the same model, JSBACH. Therefore, starting a model simulation in JSBACH-FOM with model variables created in JSBACH-PEAT is relatively simple. The only variables that change drastically in 1950 are variables related to vegetation but this is due to the start of the forestration and would happen also in an actual forest. Most importantly, the magnitude of soil carbon pools was preserved in each transition between model versions.

A table for the different scenario would be helpful to understand what the difference of the management scenarios mean. I still not understand what restoration means. Degraded peat usually mean that the peat has dried out due to drainage and the organic matter is decomposing extremely quickly.

Restoration has been explained shortly in the paragraph starting on line 63. We will modify line 194, which explains restoration in the model simulation, to clarify that here, the simulated peatland is restored to a wetland by rewetting and reintroducing wetland vegetation. It is true that the top layer has dried due to drainage. However, in boreal drained peatlands, organic matter does not decompose extremely quickly due to the low temperatures.

We will clarify in figure 1 that the restoration option will include reintroducing wetland vegetation and removal of tree cover which will hopefully make the figure and the outline of the work more easily understandable.

*Fig. 2* Here I would prefer scatter sites to see if the model has a bias or not. Additionally, I would like to see the dynamics on specific sites as well, because this would support modeled projection result of these fluxes.

We have deliberately avoided scatter sites as these, in our opinion, would give a false understanding to the reader that the model results and measurements would describe the precisely same conditions and could be compared one to one which is not the case, as explained earlier.

References are missing for the flux tower, please provide all of them, including the link.

We are unsure what references to flux towers the referee means, as we have not used any flux tower data in this paper.

It is also not mentioned which period for the measurement were taken. Is that an annual mean or sum? Please be more precise in describing the data.

We will add this information to section 2.3.

The scale of a) and b) is very different, meaning that the sink is extremely small, this reflect what I expected.

The scale is intentionally very different as the fluxes in drained forest peatland soils are indeed considerably smaller than in wetlands.

Is that really significant for all years that these sites are mainly methane sinks? Peatlands are the largest natural source of methane emissions, so I would have expected this for all sites.

The sites that indicate sinks of methane are drained forest sites on peat soils and these indeed are typically, on average, sinks of methane as explained in a previous comment.

*Environmental controls:*

It's just a suggestion, but wouldn't it be more powerful to explain the dynamics based on the measured data first . This would improve the study considerably .

The environmental controls considered in this paper have been chosen due to their known impacts on peatland methane fluxes. As this is not an empirical study, adding such analyses would make this paper likely unnecessarily long and is perhaps better to be left for another paper.

line 260 Do you mean a carbon sink of 2kg (C) /ha or Methane, you wrote C.

We meant a sink of methane which was here expressed as the amount of carbon it contains. However, we understand that this is a somewhat peculiar way to express this and will modify the units.

I could imagine that it will remain a carbon sink. Since it comes up here, I would prefer to see the full carbon dynamics, including organic carbon, carbon dioxide, and methane and importantly also the removal of biomass. As peatlands are carbon sinks, this should be reflected some how, not only in the increase of methane emissions.

The aim of this paper is to focus on the methane fluxes, not in the full carbon dynamics as including them in a sufficient detail would expand the scope of this paper. However, we do agree that these are also highly important aspects of peatland carbon balance and look forward to studying them in another paper.

*Discussion:*

line 300 : Whether or not methane becomes a source after clear-cutting depends heavily on the organic material that has been removed. Whether all or part of it is left on the land or removed has an extreme impact on carbon and methane dynamics. How was this addressed in your model and in the model cited? What does YASSO pools mean? This is an important management option that I think affects the results the most, at least in vegetation models.

We are not entirely sure what this comment means. If this refers to the biomass of the tree vegetation removed in a clear-cut, this has been taken into account by considering residual harvest material and its distribution into the different soil carbon pools as explained in section 2.1.3. The rest of the tree biomass is naturally removed from the system as it is transported for further wood production. If this refers to the soil organic carbon, this has not been removed and its removal has therefore not been considered here.

YASSO pools have been explained in section 2.1.2 and they refer to pools of organic carbon in the soil that are separated based on their different decomposition rates.

References:

Fisher, R. A., & Koven, C. D. (2020). Perspectives on the future of land surface models and the challenges of representing complex terrestrial systems. *Journal of Advances in Modeling Earth Systems*, 12, e2018MS001453. https://doi.org/10.1029/2018MS001453

Glenn, S., Heyes, A., & Moore, T. (1993). Carbon dioxide and methane fluxes from drained peat soils, southern Quebec. *Global Biogeochemical Cycles*, 7(2), 247-257. https://doi.org/10.1029/93GB00469

Hornibrook, E. R. C., Bowes, H. L., Culbert, A., & Gallego-Sala, A. V. (2009). Methanotrophy potential versus methane supply by pore water diffusion in peatlands. *Biogeosciences*, 6(8), 1491-1504. https://doi.org/10.5194/bg-6-1491-2009

Korkiakoski, M., Ojanen, P., Penttilä, T., Minkkinen, K., Sarkkola, S., Rainne, J., ... & Lohila, A. (2020). Impact of partial harvest on CH4 and N2O balances of a drained boreal peatland forest. *Agricultural and Forest Meteorology*, 295, 108168. https://doi.org/10.1016/j.agrformet.2020.108168

Li, X., Markkanen, T., Korkiakoski, M., Lohila, A., Leppänen, A., Aalto, T., ... & Raivonen, M. Modelling the alternative harvesting effects on soil CO2 and CH4 fluxes from peatland forest by JSBACH-HIMMELI model. Available at SSRN 4170450. https://dx.doi.org/10.2139/ssrn.4170450

Mäkelä, J., Knauer, J., Aurela, M., Black, A., Heimann, M., Kobayashi, H., ... & Aalto, T. (2019). Parameter calibration and stomatal conductance formulation comparison for boreal forests with adaptive population importance sampler in the land surface model JSBACH. *Geoscientific Model Development*, 12(9), 4075-4098. https://doi.org/10.5194/gmd-12-4075-2019

Ojanen, P., & Minkkinen, K. (2020). Rewetting offers rapid climate benefits for tropical and agricultural peatlands but not for forestry-drained peatlands. *Global Biogeochemical Cycles*, 34(7), e2019GB006503. https://doi.org/10.1029/2019GB006503

Raivonen, M., Smolander, S., Backman, L., Susiluoto, J., Aalto, T., Markkanen, T., ... & Vesala, T. (2017). HIMMELI v1. 0: HelsinkI Model of MEthane buiLd-up and emIssion for peatlands. *Geoscientific Model Development*, 10(12), 4665-4691. https://doi.org/10.5194/gmd-10-4665-2017

Roulet, N. T., Ash, R., Quinton, W., & Moore, T. (1993). Methane flux from drained northern peatlands: effect of a persistent water table lowering on flux. *Global Biogeochemical Cycles*, 7(4), 749-769. https://doi.org/10.1029/93GB01931

---

## Author Response (AR2)

If feasible, please make your dataset available through a public database and provide a DOI in the "Code and data availability" section of your manuscript. Please see our data policy: https://www.biogeosciences.net/policies/data_policy.html

The measurement data is owned by the co-authors and will be published separately on their forums in the context of an article related to the measurement data. Therefore we are unable to publish it in this context.

Please check figures that use both green and red/orange to see if they allow readers with color vision deficiencies to correctly interpret your findings. For more instructions please see: https://www.biogeosciences.net/submission.html#figurestables

We have rechecked the figures using online color blind tools, and as far as we can see, these figures should be readable in their current format.